# The characterization of Taklamakan dust properties using a multi-wavelength Raman polarization lidar in Kashi, China

Qiaoyun Hu[1], Haofei Wang[2,4,5], Philippe Goloub[1], Zhengqiang Li[2], Igor Veselovskii[3], Thierry Podvin[1], Kaitao Li[2], and Mikhail Korenskiy[3]

[1]Univ. Lille, CNRS, UMR 8518 - LOA - Laboratoire d'Optique Atmosphérique, F-59000 Lille, France
[2]State Environmental Protection Key Laboratory of Satellite Remote Sensing, Aerospace Information Research Institute, Chinese Academy of Sciences, Beijing, China
[3]Prokhorov General Physics Institute of the Russian Academy of Sciences, Moscow, Russia
[4]University of Chinese Academy of Sciences, Beijing, 100101, China
[5]Key Laboratory of Radiometric Calibration and Validation for Environmental Satellites, National Satellite Meteorology Center, Beijing, 100081, China

**Correspondence:** Haofei Wang (wanghf@radi.ac.cn)

**Abstract.** The Taklamakan desert is an important dust source for the global atmospheric dust budget and a cause of the dust weather in Eastern Asia. The characterization of Taklamakan dust in the source region is still very limited. To fill this gap, the DAO (Dust Aerosol Observation) was conducted in April 2019 in Kashi, China. Kashi site is about 150 km to the west rim of the Taklamakan desert and is strongly impacted by desert dust aerosols, especially in spring time, i.e. April and May. According to sun/sky photometer measurements, the aerosol optical depth (at 500 nm) varies in the range of 0.07–4.70 and the Ångström Exponent (between 440 and 870 nm) varies in the range of 0.0–0.8 in April 2019. In this study, we provide the first profiling of the $2\alpha + 3\beta + 3\delta$ parameters of Taklamakan dust based on a multi-wavelength Mie-Raman polarization lidar. For Taklamakan dust, the Ångström Exponent related to extinction coefficient (EAE, between 355 and 532 nm) is about 0.01±0.30, and the lidar ratio is found to be 45±7 (51±8–56±8) sr at 532 (355) nm. The particle linear depolarization ratios (PLDRs) are about 0.28–0.32±0.07 at 355 nm, 0.36±0.05 at 532 nm and 0.31±0.05 at 1064 nm. Both lidar ratios and depolarization ratios are higher than the typical values of Central Asia dust in the literature. The difference is probably linked to the fact that observations in the DAO campaign were collected close to the dust source, therefore, there is a large fraction of coarse-mode and giant particles (radius > 20 $\mu$m) in the Taklamakan dust. Apart from dust, fine particles coming from local anthropogenic emissions and long-range transported aerosols are also non-negligible aerosol components. The signatures of pollution emerge when dust concentration decreases. The polluted dust (defined by PLDR$_{532}$ ≤0.30 and EAE$_{355-532}$ ≥0.20) is featured with reduced PLDRs and enhanced EAE$_{355-532}$ compared with Taklamakan dust. The mean PLDRs of polluted dust generally distributed in the range of 0.20–0.30. Due to the complexity of the nature of the involved pollutants and their mixing state with dust, the lidar ratios exhibit larger variabilities compared with dust. The study provides the first reference of novel characteristics of Taklamakan dust measured by Mie-Raman polarization lidar. The data could contribute to complementing the dust model and improving the accuracy of climate modeling.

# 1 Introduction

Airborne dust is the most abundant aerosol species and accounts for nearly 35% of the total aerosol mass in the atmosphere (Boucher et al., 2013), with an annual flux of 1000–5000 Tg per year (Engelstaedter et al., 2006; Textor et al., 2006; Huneeus et al., 2011). According to the estimation of Ginoux et al. (2012), about 75% of the atmospheric dust is originated from natural emission and anthropogenic dust emission accounts for ~25%. The area spreading from the Sahara desert, the Arabian Peninsula, Central Asia to East Asia is the most significant natural dust source. Based on model simulations, Tanaka and Chiba (2006) estimated that the Saharan desert contributes to ~62% of the total dust emission and the contribution of Arabian Peninsula, Central Asia and East Asia is about half of the Saharan emission. The dust sources in North and South America, and Australia altogether account for about 25% of total emission. The suspending dust particles can directly influence the planetary radiation budget, and indirectly impact the climate through interfering with cloud properties and cloud process. Dust particles, as well as other ice nucleating particles (INP), can aide the formation of ice crystals in the heterogeneous ice nucleation regime. Due to their effective ice nucleating capability and abundant concentration, mineral dust particles are considered as the most important INP (Kanji et al., 2017). Recent studies found that atmospheric dust is also linked to the activity of tropical cyclones and rainfall (Reed et al., 2019; Thompson et al., 2019).

A comprehensive dataset of dust properties is of significant importance for understanding the effects of dust in the eco-system and for reducing the uncertainties of climate model. However, this task is very challenging and needs the support of observational data. The properties of dust particles are determined by the texture of soils, the mineralogical compositions, vegetation cover and surface properties, which could vary globally from location to location. The modeling of dust horizontal and vertical distribution, and dust cycle, i.e. dust emission, transport and deposition, is crucial to climatic modeling. So far, the vertically resolved information can only be obtained from lidar (Light detection and ranging) measurements. A multi-wavelength Mie-Raman polarization aerosol lidar can obtain multiple parameters at a vertical level. This capability makes it a useful tool for aerosol study. The profiles of backscatter coefficient, extinction coefficient and depolarization ratio derived from satellite lidar and ground-based lidars have been used as model inputs and have been proved useful for improving the accuracy of model simulation and forecasting (Yumimoto et al., 2008; Campbell et al., 2010; Sekiyama et al., 2010; Wang et al., 2013; Zhang et al., 2011, 2012).

In Asia, dust sources distribute over a large area and cover different terrain types. The high-elevated bare lands in Iran, Afghanistan and Tajikistan, and the Taklamakan desert in the Tarim basin, the Loess plateau and the Gobi desert in China are the main dust sources. In addition, excessive land-use and human activities formed new dust sources. There are a good number of publications reporting transported Asian dust observed in the downwind countries in East Asian (Liu et al., 2002; Murayama et al., 2004; Huang et al., 2008; Iwasaka et al., 2008). Long-range transported dust can cross the Pacific ocean and occasionally reach America (VanCuren and Cahill, 2002; Uno et al., 2009). However, very few field campaigns have been carried out for Asian dust study. Compared with Saharan dust, the characteristics of Asian dust were not adequately explored. The earliest field campaign characterizing Asian dust date back to 1989, when an experiment was carried out in Tajikistan for studying desert dust properties and the impact on meteorological conditions. The CADEX (Central Asian Dust EXperiment)

project was planned to provide a data set of optical and microphysical properties of dust from Central Asia. A multi-wavelength Mie-Raman polarization lidar was deployed in Dushanbe, Tajikistan. This results in Hofer et al. (2017) and Hofer et al. (2020) provided important dust properties, such as vertically resolved lidar ratios, linear depolarization ratios and mass concentrations. In 2002 and 2009, a elastic polarization lidar system (without Raman channel) was set up in Aksu (40.62°N, 80.83°E, in Xinjiang, China) near the north rim of Taklamakan desert (Kai et al., 2008; Jin et al., 2010). Jin et al. (2010) obtained the first lidar ratio of the Taklamakan dust in the source region, however, it requires extra assumptions and supplementary measurements. Sparse lidar observations in the downwind of transported Taklamakan dust have been reported but none of them provides intensive dust characteristics and the observation sites are far from the desert.

In 2019, the DAO (Dust Aerosol Observation) campaign was conducted in April to June in China. This campaign was supported by the "Belt and Road Initiative" and involved researchers from China, France and Russia. The first observation site in the DAO campaign is in Kashi (also called Kashgar) in April 2019, which is about 150 km to the western rim of the Taklamakan desert. The objective of the first session of DAO campaign is to study the characteristics of Taklamakan dust. The second session of the campaign was in Beijing in May and June, for investigating the impact of transported dust on the air quality in megacity. The main topic of this paper is the characterization of Taklamakan dust, therefore, only the measurements in Kashi will be analyzed. This study is organized into 5 sections. The description of DAO campaign is presented in Section 2, and the results and case study is in Section 3. The discussions and conclusions are presented in Section 4 and 5, respectively.

## 2    The DAO (Dust Aerosol Observation) campaign

### 2.1    Overview

The Taklamakan desert is located in the center of the Tarim Basin in the Uygur Autonomous Region of Xinjiang, China, covering an area of about 320,000 km$^2$. The mean elevation of the Taklamakan desert is about 1200–1500 m a.s.l (Petrov and S.Alitto, 2019). It is surrounded in three directions by high mountain ranges (see Figure 1). The observation site (39.51°N, 75.93°E, time zone: GMT+08:00) is in the northwest of the Kashi city and close to the border to Tajikistan, Kyrgyzstan and Afghanistan. Kashi features a desert climate with a big temperature difference between winter and summer. The coldest month is January with average temperature of -10.2–0.3°C and the warmest month is in July with average temperature of 18.6–32.1°C. The annual rainfall in Kashi is about 64 mm. The spring in Kashi is long and comes quickly. The rapidly heated surface sand in the desert could generate ascending currents which could result in frequent dust storm in springtime. This is the main reason that the field campaign was performed in springtime.

Except for desert dust, anthropogenic emission is another important aerosol source. There are about 4.65 million habitants (predicted for 2017, see the link) in the Kashi prefecture, including the Kashi city and 11 subordinate counties. Kashi prefecture is a very populated region in Xinjiang with more than 1000 persons per square kilometer in the city center (Doxsey-Whitfield et al., 2015). Fine aerosol particles originated from biomass burning and local anthropogenic emissions, such as heating, traffic and industrial pollution are an important aerosol component. Moreover, there are populated cities in the neighboring countries

such as Kyrgyzstan and Tajikistan. Under favorable meteorological conditions, various aerosols, for example, pollution, could be potentially transported to Kashi and mix with dust aerosols.

## 2.2 Instrumentation and methodology

**Lidar system**

The multi-wavelength Mie-Raman polarization lidar called LILAS (Lille Lidar Atmosphere Study) is the main instrument installed in observation site. The lidar system has been operated in LOA (Laboratoire d'Optique Atmosphérique, Lille, France) since 2013 (Bovchaliuk et al., 2016; Veselovskii et al., 2016; Hu et al., 2019). During the DAO campaign, LILAS was transported from Lille to Kashi (and Beijing in the second session of the campaign) to perform observations. LILAS uses a Nd: YAG laser that emits at three wavelengths: 355, 532 and 1064 nm. The laser repetition rate is 20 Hz. A Glan prism is used to clean the polarization of the laser beam. The emitting power after the Glan prism is about 70, 90 and 100 mJ at 355, 532 and 1064 nm, respectively. LILAS system has three Raman channels, including 387 (vibrational-rotational), 530 (rotational) and 408 nm (water vapor). The use of rotational Raman at 530 nm provides a stronger Raman signal and relieves the dependence of the derived extinction and backscatter coefficients on the assumption of Ångström exponent (Veselovskii et al., 2015). The backscattered light is collected by a 400 mm Newton telescope. The incomplete overlap range of LILAS system is about 1000–1500 m in distance, depending on the selected field of view angle. In the receiving optics, the three elastic channels are equipped with both a perpendicular and a parallel channel with respect to the polarization plane of the emitted linearly polarized laser light, in order to measure the linear depolarization ratio at three wavelengths. LILAS can provide the profiles of the $2\alpha+3\beta+3\delta$ ($\alpha$: extinction coefficient, $\beta$: backscatter coefficient, $\delta$: particle linear depolarization ratio (PLDR)) parameters. Benefited from the coupled Raman channels, the extinction and backscatter coefficients at 355 and 532 nm are calculated using the Raman method proposed by Ansmann et al. (1992). The Raman signal generated by the radiation at 1064 nm is not measured by LILAS, thereby Raman method is not applicable. The backscatter coefficient at 1064 nm is calculated using the Klett method, where a vertically constant lidar ratio (extinction-to-backscatter ratio) is assumed (Klett, 1985). The particle linear depolarization ratios are derived from Equation 1:

$$\delta^p = \frac{(1+\delta^m)\delta^v R - (1+\delta^v)\delta^m}{(1+\delta^m)R - (1+\delta^v)}, \tag{1}$$

where $R$ represents the ratio of the total backscatter coefficient, involving molecules and particles, to the particle backscatter coefficient. $\delta^m$ represents the molecular depolarization ratio. $\delta^v$ represents the volume linear depolarization ratio (VLDR), which equals to the calibration coefficient multiplied by the ratio of the signal of the perpendicular channel to the parallel channel. The polarization calibration is performed following the $\pm45°$ method (Freudenthaler et al., 2009). During the DAO campaign, the polarization calibration has been performed at least once per day.

The Ångström exponent of the extinction coefficient and backscatter coefficient are calculated by the Equation 2:

$$\mathring{A} = -\frac{\log p(\lambda_1) - \log p(\lambda_2)}{\log \lambda_1 - \log \lambda_2} \tag{2}$$

where $p(\lambda)$ represents the optical parameters, such as AOD, extinction or backscatter coefficient at wavelength $\lambda$, $\mathring{A}$ represents the Ångström exponent of the corresponding parameters $p(\lambda)$. The statistical error of lidar derived parameters is estimated using the method presented in Hu et al. (2019). The data presented in this study are recorded in nighttime, so the background radiation is negligible. The error in the extinction and backscatter coefficient is about 10%, which leads to about 15% of error in the lidar ratios, at 355 and 532 nm. The error in the backscatter coefficient at 1064 nm is about 20%. The error in PLDR is calculated in terms of the backscatter ratio, VLDR and molecular depolarization ratio. For the data presented in this study, the error in PLDR is no greater than 15% at 532 and 1064 nm. Therefore, we conservatively use 15% as the error in PLDR for 532 and 1064 nm. At 355 nm, the error of 15% still holds when dust concentration is high enough, but when the concentration drops, the error could exceed 15%. In the case study, errors at 355 nm are calculated separately. The errors for the water vapor mixing ratio (WVMR) and relative humidity (RH) are about 20%.

**Sun/sky photometer**

Three sun/sky photometers are deployed in the Kashi observation site. One is affiliated to the AERONET (AErosol RObotic NETwork, Holben et al. (1998)) network and the other two are affiliated to SONET (Sun-Sky Radiometer Observation Network). SONET is a ground-based sun/photometer network with the extension of multi-wavelength polarization measurement capability to provide long-term columnar atmospheric aerosol properties over China (Li et al., 2018). The three sun/sky photometers provide complementary measurements by following different measurement protocols. In all, they can measure daytime aerosol optical depth (denoted as AOD hereafter) at 340, 380, 440, 675, 870, 1020 and 1640 nm, polarized/unpolarized sky radiances at 440, 675, 870 and 1020 nm and moon AOD as well. The succeeding data treatment and retrieval are performed following the protocols and standards of AERONET or SONET, depending on the affiliation of the instruments.

**Satellite data**

Satellite data have complementary advantages due to their large spatial coverage compared with ground-based remote sensing technique. In order to monitor dust activities of the Taklamakan desert, we use the UV aerosol index (UVAI hereafter) derived from the OMPS (Ozone Mapping Profiler Suite) onboard the Suomi-NPP (National Polar-orbiting Partnership) satellite (Flynn et al., 2004; Seftor et al., 2014). OMPS provides full daily coverage data and the overpass time for Kashi region is around 06:30 UTC. The UVAI is calculated using the signal in the 340 and 380 nm channels (Hsu et al., 1999):

$$\text{UVAI} = -100 \times \left\{ \log_{10} \left[ \frac{I_{340}}{I_{380}} \right]_{meas} - \log_{10} \left[ \frac{I_{340}}{I_{380}} \right]_{calc} \right\}, \tag{3}$$

where $I_{340}$ and $I_{380}$ represent the backscattered radiance at 340 and 380 nm. The subscripts "meas" and "calc" respectively represent the real measurements and model simulation in a pure Rayleigh atmosphere. By the definition of UVAI, its positive values correspond to UV-absorptive aerosols such as desert dust and carbonaceous aerosols. Hence, the UVAI from OMPS is a good parameter for monitoring the activity of the Taklamakan desert.

**Auxiliary data**

A radiosonde station (39.47°N, 75.99°N) in Kashi is 6 km to the observation site. The data are accessible on the website of the Wyoming weather data website (see the link). The radio sounding data are recorded at 00:00 and 12:00 every day at local time.

They provide the vertical temperature and pressure profiles for the calculation of molecule scattering parameters in lidar data processing. The HYSPLIT (Hybrid Single-Particle Lagrangian Integrated Trajectory, Stein et al. (2015); Rolph et al. (2017)) model developed by the National Oceanic and Atmospheric Administration (NOAA) Air Resources Laboratory is used for the back trajectory of the air mass and for the air mass clustering. The HYSPLIT model is driven by the 0.5° gridded GDAS (Global Data Assimilation System) data and could produce the transport pathways of the air mass at different vertical levels. Besides, instruments measuring particulate matter (PM10 and PM2.5), gas concentration (SO2, O3 and NOx), particle size distribution, particle scattering and absorption coefficients, solar radiation and a cloud monitor are also deployed in the field campaign. These data contribute to relevant air quality and solar radiation studies within the frame of the DAO campaign.

## 3  Results and analysis

### 3.1  Overview

Figure 2 presents the monthly averaged AOD at 500 nm, Ångström exponent between 440 and 870 nm and the FMF (fine mode fraction, the fraction of fine mode AOD to total AOD) in Kashi site from 2013 to 2017. The data are derived from SONET network. The highest AOD occurs in spring, i.e. March and April, while the lowest values occur in summer time, i.e. June and July. The Ångström exponent is positively correlated to the FMF and negatively correlated to the AOD. The lowest mean Ångstöm exponent occurs in March and April, indicating that dust particles are dominant due to the seasonal increase of dust activities in this period (Littmann, 1991; Qian et al., 2002). In December and January, the Ångström exponent and FMF increase significantly, which proves that fine particles are an important aerosol component in winter.

Figure 3(a) plots the AOD at 500 nm and the Ångström exponent measured during the DAO campaign in April 2019. The AOD varies from 0.07 to 4.70 and the Ångström exponent varies from 0.0 to 0.8. For AOD greater than 0.2, the corresponding Ångström exponent mostly falls into the range of 0.0 to 0.2. While for AOD lower than 0.2, the Ångström exponent is mostly between 0.3 to 0.7. The negative correlation between the AOD and the Ångström exponent indicates that coarse particles are the main cause for the increase of AOD. This argument is supported by the variation of the particulate matter plotted in Figure 3(b).

We select four representative cases from the nearly 1 month lidar observations. The four cases are recorded on 03, 09, 15 and 24 April 2019. In order to distinguish "pure" Taklamakan dust observations, we define Taklamakan dust by $EAE_{355-532}$ (Ångström exponent related to extinction coefficient) smaller than 0.1 and $PLDR_{532}$ greater than 0.32 at 532 nm. Polluted dust is defined with PLDR smaller than 0.30 at 532 nm and EAE no smaller than 0.2. Back trajectories are also used as a reference for identifying the aerosol origins. The maps of UVAI are plotted in Figure 4. On 09 and 24 April, intense aerosol plumes were observed over the Taklamakan desert. One extreme dust event occurred on 24 April when the AOD (at 500 nm) reached about 4.70 at 08:40 UTC, with instantaneous Ångström exponent about -0.02 and the visibility about 1 km (see the link). The PM10 increased to the monthly maximum on 24 April, reaching nearly 1500 $\mu$g/m$^3$. It should be noted that, in this extreme case where AOD reached 4.70, the accuracy of the measured AOD may degrade because of the decreasing signal-to-noise ratio. Moreover, weak incoming solar radiation might disturb the performance of the cloud screening in the quality control

procedure, thus disabling the discrimination of cloud contaminated and non-contaminated measurements. On 03 and 15 April, the activity of the Taklamakan desert became less intense compared with the first two cases. The concentration of dust particles decreased and the features of polluted dust appeared. Lidar quicklooks at 532 nm for the four cases are plotted in Figure 5.

## 3.2 Case studies

### 3.2.1 Case 1: 09 April 2019

Dust plumes over the Taklamakan desert are detected on 07, 08 and 09 April, as shown in Figure 4. The most intense plume in the three days appeared on 07 April, with maximum UVAI about 4.0. On 09 April, a belt-like plume appeared in the north and northwest of the desert. Figure 5(b) shows the range-corrected lidar signal at 532 nm collected between 9 and 10 April. The boundary layer height slightly increases from 3000 m to 4000 m in the night, and strong backscattered lidar signal is seen below 2000 m. Figure 6 shows the profiles of the optical properties, WVMR and RH averaged between 17:00 and 22:00 UTC, 09 April 2019. The extinction coefficients gradually decrease with height. At 1000 m, the extinction coefficients are greater than 0.5 km$^{-1}$ and remain almost stable below 2000 m. The RH is no more than 40±8% below 2000 m and rises to 60±12% at 3800 m. The lidar ratio varies between 40±6 and 48±7 sr at 532 nm and between 55±8 and 62±9 sr at 355 nm. The PLDR is about 0.32±0.05 at 355 nm and 0.36±0.05 at 532 nm. The VLDR at 1064 nm is about 0.32±0.03. The backscatter coefficient, as well as the PLDR at 1064 nm is not available above 1800 m, since the 1064 nm lidar signal has distorted in upper boundary layer. We can expect that the VLDR is approximate to PLDR at 1064 nm under in this case, because the dust content is so high that molecular scattering at 1064 nm can be neglected. The EAE$_{355-532}$ is about -0.10±0.30 at 800 m and rises to 0.10±0.30 at 3800 m. The BAE$_{355-532}$ (Ångström exponent related to backscatter coefficient) is negative and varies from -0.7±0.3 to -0.4±0.3. Below 3000 m, the lidar ratios mildly decrease with height, while the PLDRs do not show obvious vertical variations. Above 3000 m, the vertical variations in the lidar ratios and PLDRs become more significant. The vertical variations of the lidar ratios and PLDRs are possibly the result of particle sedimentation or/and vertically dependent particle origins.

On 09 April, the Taklamakan desert was covered by a low-pressure zone with easterly and northeasterly wind prevailing over the western part of the desert. It is a favorable condition for the elevation of dust particles. Figure 7 shows the 48-hour back trajectory ending at 20:00 UTC for air mass at 1000, 2000 and 3000 m. The air masses at the three vertical levels are originated from the Taklamakan desert. They all passed over the area where dust plumes have been observed and then diverged when approaching the rim of the desert. In the end, the air masses at 1000, 2000 and 3000 m arrived at the observation site from the northeast, east and southeast respectively. The particles observed by LILAS on 09 April are fresh desert dust, without long-range transport. Therefore, they could contain a large fraction of coarse-mode particles especially giant particles (radius> 20 $\mu$m). Moreover, the back trajectories in Figure 7 shows convective strong air flows arising from below 500 m to 3000 m within 3 hours, suggesting the possibility of lifting large particles near the surface to higher levels.

### 3.2.2 Case 2: 24 April 2019

On 24 April, the observation site was enclosed by floating dust. In the daytime, the sky radiance dropped below the detection limit of the sun/sky photometer, so the AERONET and SONET retrieval can not be applied. A large and intense plume was first detected in the morning of 23 April 2019 (Figure 4). On 24 April, a hot spot of UVAI appeared over the observation site. The daily average of AOD is 3.63 and Ångström exponent is about -0.01, according to the daytime sun/sky photometer measurements. The lidar quicklook on 24 April in Figure 5 shows that the boundary layer height rises from about 1200 m to 2000 m from 14:00 to 24:00 UTC. Due to the high dust attenuation in the boundary layer, both sun/sky photometer and lidar cannot detect whether clouds exist on 24 April. Figure 8 plots the averaged parameters between 15:00–24:00 UTC, 24 April 2019. The dust layer was so thick that the laser beam can not penetrate. The amplitude of Raman signal dropped by 5–6 orders in the lower 2000 m. In this condition, we can not find an aerosol-free zone to for the calibration of lidar signal, therefore, the calculation of the backscatter coefficient using Raman method is not possible. But the extinction coefficient can be derived from the Raman signal (Ansmann et al., 1992). The extinction coefficients are $1.0\pm0.1$ km$^{-1}$ at 800 m and increases to about $1.5\pm0.2$ km $^{-1}$ at 1500 m. The extinction coefficient at 355 nm is removed at above 1500 m because it starts to oscillate due to insufficient signal-to-noise ratio. The extinction coefficient at 532 nm decreases to about $1.1\pm0.1$ km$^{-1}$ at 2000 m. By assuming that the lidar ratios are about 55 sr and 45 sr at 355 and 532 nm, respectively, we obtain the backscatter coefficient from the extinction coefficient, and then calculate the PLDRs (in Figure 8(c)). The PLDR is about 0.32 at 355 nm and 0.37 at 532 nm, which are rather consistent with the results in Case 1. The uncertainties of the PLDRs are not accessible because the uncertainties of the assumption of lidar ratio are not known.

The back trajectories (not shown) indicates that dust particles (at 1000 and 2000 m) are originated from the northeast and east, where intense dust plumes were observed on 23 and 24 April. Figure 9 shows synoptic conditions at 00:00 UTC, 23 April and 06:00 UTC, 24 April. The meteorological conditions on 23 and 24 April are favorable for dust emission, similar to Case 1. The Taklamakan desert is enclosed by a low-pressure zone (Figs. 9(a) and (c)). The plume observed by OMPS on 23 April was probably lofted in the local morning. In the eastern part of the Taklamakan desert, 37–39°N, 83–88°E, the wind velocity at 10 m (a.g.l) reaches more than 50 km/h (Figure. 9(a)) and at 850 hPa level the maximum wind velocity reaches 90 km/h (Figure. 9(b)). The high wind velocity near the surface and large vertical wind gradient help elevate dust particles from the surface into the atmosphere. On 23 and 24 April, easterly and northeasterly wind are prevailing in the desert region, thus blowing the lifted dust particles to the observation site. Case 2 is a more severe manifestation of Case 1 regarding the intensity of the dust loading. In both cases, the observed dust particles are originated from nearby dust source. Compared with typical depolarization ratios in worldwide dust observations, which are 0.23-0.30 at 355 nm and 0.30-0.35 at 532 nm, the depolarization ratios we obtained in the two cases are relatively higher (Veselovskii et al., 2016; Freudenthaler et al., 2009; Hofer et al., 2017, 2020). While previous observation sites were mostly not as close to the dust source as in our campaign, the differences are probably due to the fraction of coarse-mode particles that remain in our dust observation. Burton et al. (2015) also observed dust particles near the source in North America and reported PLDR of 0.37 at 532 nm, which is consistent with our result. However, the PLDR at 355 nm measured by Burton et al. (2015) is about 0.24, lower than what we obtained.

### 3.2.3 Case 3: 15 April 2019

On 15 April, the daily mean AOD on 15 April was 0.63 and the Ångström exponent was about 0.10. Compared with the previous two cases, the Ångström exponent increase obviously. The boundary layer height started to increase at 15:00 UTC and stayed at 3500 m in the night of 15 April. Cirrus clouds were continuously present during the period of lidar measurement (Figure 5(c)). The lidar derived profiles between 18:00–20:00 UTC are plotted in Figure 10. The extinction coefficients in the boundary layer are about 0.15 $km^{-1}$ and decrease to almost zero at 3500 m. The RH increases up to 60±12% at 2200 m. Below this height, the lidar ratio, PLDR, EAE and BAE are almost stable. The lidar ratio is about 51±8 sr at 355 nm and 45±7 sr at 532 nm. The PLDRs at 355, 532 and 1064 nm are around 0.32±0.07, 0.34±0.05 and 0.31±0.05, respectively. The $EAE_{355-532}$ is about 0.02±0.30, showing a gentle increase with height, and the $BAE_{355-532}$ is about -0.29±0.30. Above 2200 m, the RH starts to increase and reaches its maximum, i.e. 80±16%, at 2800 m. The $EAE_{355-532}$ and $BAE_{355-532}$ increase to 0.10±0.30 and -0.06±0.30, respectively. On contrary, the lidar ratios and PLDRs decrease and reach their minima at about 2800 m. The lidar ratio is about 40±6 sr at 2400–2800 m, with a weak spectral dependence, and the PLDRs are about 0.23±0.06 at 355 nm, 0.26±0.04 at 532 nm and 0.24±0.03 at 1064 nm. It should be noticed that the backscatter coefficient at 1064 nm is performed using Klett method with an assumption of lidar ratio equal to 40 sr (Klett, 1985).

Dust activities were observed by the OMPS on 13 and 15 April 2019 (Figure 4), while the intensity was less stronger than in Case 1 and 2 and the distance between the dust plume and the observation site is farther. The 48-hour back trajectories ending at 19:00 UTC are shown in Figure 11. Air masses at the three vertical levels (1000, 2000 and 3000 m) are originated from the eastern part of the Taklamakan desert, where no intense dust activities have been observed by OMPS in the recent three days. It explains the decrease of dust content in the boundary layer. When dust loading decreases, the impact of fine mode particles emerges. The changes of EAE, BAE, PLDR and lidar ratios above 2200 m are a clear evidence of polluted dust. The pollution could be lifted up from the ground in local area by convection or be transported from other area. Additionally, the RH at above 2500 m is about 60±12%–80±16%, which could lead to the hygroscopic growth of some aerosol species. Pure dust is regarded as hydrophobic aerosols because its compounds are insoluble, but when mixed with hygroscopic aerosol species, for example, nitrate, the ensemble of aerosol mixture could become hygroscopic. The fine mode particles can be hydrophobic or hydroscopic, depending on their chemical compositions(Carrico et al., 2003; Shi et al., 2008; Pan et al., 2009). In this case, there were no clear evidence indicating the occurrence of hygroscopic growth or the mixing state of dust and pollution particles.

### 3.2.4 Case 4: 03 April 2019

The daily mean AOD on 03 April is 0.16 and the Ångström exponent is about 0.11. The boundary layer height is about 3000 to 4000 m, rising slightly in the night of 03–04 April. Starting from 16:30 UTC, some liquid cloud layers occurred at the top of the boundary layer (Figure 5(a)). Figure 12 shows the profiles derived from lidar observations at 14:00–16:00 UTC, 03 April. The extinction coefficients decrease from about 0.28±0.03 $km^{-1}$ at 1000 m to about 0.10±0.01 $km^{-1}$ at 3000 m, with $EAE_{355-532}$ ($BAE_{355-532}$) increasing from 0.01±0.30 (-0.38±0.3) to 0.28±0.30 (0.02±0.30). Below 2100 m, the lidar ratios are about 45±7 sr at 532 nm and 51±8 sr at 355 nm. The PLDRs are about 0.35±0.05 at 532 nm and 0.32±0.05 at 1064 nm

and 0.28±–0.32±0.07 at 355 nm. Between 2100 and 3000 m, the variation of lidar ratios is not monotonic. At 2500 m, the lidar ratios reach the minimum of $38 \pm 6$ sr at 532 nm and $42 \pm 6$ sr at 355 nm, and the PLDRs are about 0.27±0.06 at 355 nm and 0.33±0.05 at 532 nm. Both the lidar ratios and PLDRs at 2400-2800 m range are very consistent with the properties of Central Asia dust reported by Hofer et al. (2017) and Hofer et al. (2020). Above 2500 m, the lidar ratios and the RH (as well as WVMR) re-increase, and PLDRs decrease. At 3000 m, the PLDR at 532 nm drops below 0.30, suggesting that aerosols are different from those at lower boundary layer. The signatures of lidar ratio, RH and PLDR are possibly linked to the occurrence of polluted dust particles. In addition, long-range transported dust could also possess such PLDRs due to the deposition of big particles in the transport. This case is classified as polluted dust because the PLDR below 0.30 at 532 nm, the increase of WVMR (as well as RH) and $EAE_{355-532}$) at the boundary layer top fit better the characteristics of polluted dust.

Figure 13 plots the 72-hour back trajectories for 1000, 2000 and 3000 m. Air masses at 1000 and 2000 m are from the Taklamakan desert, while the air mass at 3000 m is from Central Asia. It corroborates the similarities of the lidar ratios and PLDRs between the measurements in Hofer's studies and in our study. When extending the trajectory duration to 96 hours, the results (not plotted) suggest that air mass at 3000 m is originated from North Africa. This result suggests that dust particles observed in Kashi may have a long-transported aerosol component. The air mass clustering based on 24-hour back trajectories (not shown) indicates that about 52% of air mass at 3000 m is from North Africa and Arabian Peninsula. At 3500 m, this proportion increases to 74% and there is also a fraction of air mass coming from Europe. The complexity in the aerosol sources in the transport pathways explains the variability of aerosol properties at upper boundary layer in Case 4.

## 4 Discussion

**Aerosol source**

The optical parameters in the 4 cases are summarized in Table 2. During the campaign, dust is undoubtedly the predominant component. In dust events (Case 1 and Case 2), dust particles are lifted from the Taklamakan desert by the low-pressure system along with strong wind, and then blown to the observation site by the easterly or northeasterly wind. In dry deposition, coarse-mode particles, especially giant particles settle down faster than the fine-mode dust particles. In many previous campaigns, the observed dust particles have undergone long-range transport, ranging from several hundreds or thousands kilometers (Dieudonné et al., 2015; Murayama et al., 2004; Veselovskii et al., 2016; Ansmann et al., 2003; Haarig et al., 2017; Hofer et al., 2017, 2020; Filioglou et al., 2020). While the transport distance is much shorter in DAO campaign. Thus, the observed dust particles in DAO campaign are more likely to contain a large fraction of coarse-mode and giant particles, which is an important difference of our observations compared with most previous observations. Moreover, the mineral composition of dust is size-dependent. In the study of Saharan dust, Kandler et al. (2009, 2011) found a tendency of higher quartz content in larger particles, but a significant fraction of sulfate was found in the size range smaller than 1 $\mu$m. The iron-bearing minerals, which is linked to the dust absorption, are more concentrated in the fraction with radius smaller than 2.0 $\mu$m. The difference in the size distribution could lead to a difference in mineralogical composition and chemical properties (Ryder et al., 2018; Biagio et al., 2019).

The influence of pollution is not clearly seen in the dust storms. However, when the activities of the Taklamakan desert wane and dust concentration becomes lower, the impact of pollution emerges. Observations in Case 3 clearly demonstrate the contrast of dust in the lower boundary layer and polluted dust particles at the boundary layer top. During the 1-month campaign, the traces of pollution, featured with increased $EAE_{355-532}$ and decreased PLDRs are frequently observed. The evidence of pollution in Taklamakan dust has been found in previous in-situ measurements. Huang et al. (2010) sampled aerosol particles

in springtime at Tazhong site, which is located in the north rim of Taklamakan desert, and found that the As element was moderately enriched. The As element is a tracer of pollution, originated probably from coal burning. It is also found that the concentration of sulfate in Taklamakan dust is at a high level. The increased concentration of sulfate in the Taklamakan dust could be related to the provenance of the Taklamakan desert, because it is speculated to be ocean 5–7 millions years ago (Sun and Liu, 2006). Sulfate could also come from anthropogenic emission, for example, the uptake of the SO2 gases. Iwasaka et al.

(2003) examined the aerosol samples using electron microscopy in Dunhuang, China, which is in the downwind of transported Taklamakan dust. They found that mineral dust is the main component in the coarse-mode aerosols, while in the fine mode, ammonia sulfate, which is mainly from anthropogenic emissions, is the major component. These studies indicate that the Taklamakan dust near the source region have been contaminated by other aerosols with anthropogenic origins. It is in agreement with our analysis, however, in this study we cannot clarify the exact involved aerosol species and the mixing state in the pol-

luted dust. In our study, polluted dust mostly appeared at the boundary layer top, which agrees with the finding of Iwasaka et al. (2003). These fine particles are possibly lifted by convective air flow and then remain at higher altitude as bigger particles settle down.

Long-range transported aerosols are another possible aerosol origin in Kashi. Based on model simulations, some previous studies have reported intercontinental dust transport from North Africa or the Middle East to the East Asia (Park et al., 2005;

Tanaka et al., 2005; Sugimoto et al., 2019). Figure 14 plots the air mass clustering for three different vertical levels in April 2019. The contribution of air masses from Central Asia, Middle East, Europe and North Africa always exists and the influence increases with height. At 1000 m, the main aerosol source is from the Taklamakan desert and accounts for about 73%. At 3000 m, air mass from Central Asia, Middle East and North Africa accounts for about 51%, and there is about 2% of air mass from Europe. While at 4000 m, air mass from the Taklamakan occupies only 29% and the rest are from Central Asia, the Middle

East and North Africa. The west-to-east airmass transport is associated with the midlatitude westerlies, which is a continuous force for air mass transport (Yumimoto et al., 2009; Yu et al., 2019). In Case 4, we observed dust signatures that differ from Taklamakan dust but correspond well with the results from Hofer et al. (2017) and Hofer et al. (2020) in Dushanbe. Nevertheless, there are various aerosol sources in the intercontinental transport pathway, such as dust from North Africa, Middle East and Central Asia, pollution and biomass burning from East Europe. Moreover, the aerosol properties could be modified during

the transport. Hence, it is difficult to find out the exact aerosol types using lidar observations.

**Lidar ratio and depolarization ratio**

We found that, for Taklamakan dust, the lidar ratios are about 45±7 sr at 532 nm and 51–56±8 sr at 355 nm. The PLDRs are about 0.28–0.32±0.07 at 355 nm, 0.36±0.05 at 532 nm and 0.31±0.05 at 1064 nm. Table 3 presents an overview of the lidar

ratios and PLDRs of Asian, Saharan and American dust in previous publications. Jin et al. (2010) derived a lidar ratio of 42±3

sr at 532 nm for Taklamakan dust, which agrees well with our results. The observation site in Jin et al. (2010) was very close
to Kashi, however, their results were based on the observations of an elastic lidar, so it requires the assumption of vertically
independent lidar ratio and complementary measurements. Observations obtained from other Asian sites show mean lidar ratio
in the range of 40–50 (39–43 ) sr and PLDR in the range of 0.17–0.29 (0.20–0.35) at 355(532) nm (Dieudonné et al., 2015;
Murayama et al., 2004; Hofer et al., 2017, 2020; Filioglou et al., 2020). Both the lidar ratios and PLDRs are slightly lower than
the results we obtained from Taklamakan dust. Case 4 in this study shows the coincidence of characteristics of Taklamakan dust
in the lower boundary layer and Central Asian dust (Hofer et al., 2017) in the upper boundary layer. This coincidence proves
that the differences of lidar ratios and PLDRs between Taklamakan dust and Central Asia dust are not caused by a systematic
bias of measurements in two different lidar systems, but that the two types of dust are optically different.

The large fraction of coarse-mode and giant particles in Taklamakan dust are supposed to be the main reason responsible
for this difference. Moreover, differences of the dust mineralogical composition in various geographical locations may also
contribute to the differences in optical properties. For example, observations in SAMUM and SHADOW campaigns revealed
lower PLDRs and higher lidar ratios in Saharan dust compared with Asian dust (Groß et al., 2011; Veselovskii et al., 2016,
2020). It could be explained by the argument that Saharan dusts tend to be more absorbing than Asian dust due to its relative
higher content of iron oxides (Biagio et al., 2019).

Recent studies concluded that there are similarities in dust size and shape parameters, which explain the relatively uniform
distribution dust PLDRs for globally distributed dust sources. Nevertheless, the variability of dust properties should still be
considered. PLDRs as high as Taklamakan dust have ever been found in several previous studies. Burton et al. (2015) found
comparable PLDR of 0.37 at 532 nm in American dust near the source, but at 355 nm, the PLDR was about 0.24, falling in
the typical range of dust. Sakai et al. (2010) derived PLDR (at 532 nm) of 0.39 for Asian and Saharan dust with high number
concentration of supermicrometer particles, while for submicrometer particles, the PLDR was about 0.14–0.17. It proves that
increase of big particle concentration could strongly increase the PLDR. Miffre et al. (2016) measured artificial dust samples
with mainly submicrometer particles and derived PLDR of 0.37 at 355 nm and 0.36 at 532 nm. The high PLDRs are likely
caused by the sharp edges and corners produced in the fabrication of dust samples. In naturally formed dust particles, these
corners or edges may be trimmed by aeolian or fluvial erosion. This could be one reason why previous dust observations never
found PLDR greater than 0.30 at 355 nm. While near the source and in heavy dust event, we suppose that the lifted dust may
contain a fraction of big and morphologically complicated particles, which have strong depolarizing effects. Since Taklamakan
dust observations in the source region are quite rare, more observational data are needed for complementing the data set of dust
characteristics.

## 5 Conclusions

The first session of DAO campaign was conducted in Kashi, China in April 2019. The objective of DAO campaign is to provide a comprehensive characterization of Taklamakan dust using multi-wavelength Mie-Raman lidar measurements. During the nearly 1 month campaign, we found that, dust particles, originated mainly from the Taklamakan desert, are the dominant aerosol component in springtime in Kashi, while the influence of fine-mode particles needs also to be considered. Kashi is a populated region, pollution emitted from anthropogenic activities very likely the main component in fine-mode aerosols. Additionally, air mass clustering using the HYSPLIT model suggests that long-range transported aerosols from Africa, Europe, the Middle East and Central Asia could be a potential aerosol origin in Kashi. This study provides the first characterization of the spectral lidar ratios and PLDRs of the Taklamakan dust. One distinct feature of Taklamakan dust is its relatively high PLDRs compared with other Asian dust and Saharan dust. We suppose this difference is related to the coarse-mode and giant particles that remain in the Taklamakan dust near the source region. The results fill the gap of the characterization of Taklamakan dust and provide reference for succeeding studies and for implementing the climate modeling. This study also points out the importance of considering the dust mixing with pollution in climate modeling. Our results show that, in the most dusty season of the year and at an observation site with 150 km to the desert, the observed Taklamakan dust has already been polluted. Pollution could alter the optical and microphysical properties of dust particles, thus influencing the direct radiative forcing. Moreover, polluted dust could modify the cloud formation process by acting as cloud condensation nuclei and ice nuclei, which impose indirect influence on the earth's radiation budget and the long-term climate change. The DAO campaign offers a nice collection of measurements relevant to cloud-dust interactions, which will be presented in the next step.

*Data availability.* The satellite data from OMPS and AIRS can be found in NASA's GES DIS service center. The meteorological data, GDAS data and the HYSPLIT dispersion model are available in the NOAA ARL site (https://ready.arl.noaa.gov/HYSPLIT.php). All the other data presented in this study are available upon any request of readers.

*Author contributions.* The project was supervised by PG and ZL. QH, TP and IV were in charge of the Lidar operation and maintenance. QH, IV and HW performed the data analysis. QH wrote the manuscript of this paper. KL provide the sun/sky photometer data. MK helped in the lidar operation and instrument preparation.

*Competing interests.* The authors declare that they have no conflict of interest.

*Acknowledgements.* We acknowledge the colleagues in the Institute of Remote Sensing and Digital Earth for their kind help and for the financial support. And we thank the colleagues in the GPI RAS (Prokhorov General Physics Institute of the Russian Academy of Sciences),

Service National d'Observation PHOTONS/AERONET-EARLINET, component of ACTRIS infrastructure, ESA/IDEAS+ and labex CaPPA projects, IAP (Institute of Atmospheric Physics) for the participation and the efforts they have made for the campaign. The Russian Science Foundation (project 16-17-10241) is acknowledged for providing lidar data processing algorithms.

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

**Table 1.** Daily averaged AOD at 500 nm and Ångström exponent (between 440 and 870 nm) measured by the sun/sky photometer in daytime. The values on the right side of '$\pm$' represent the standard deviation of the values on the left side.

|  | $AOD_{500}$ | $AE_{440-870}$ |
|---|---|---|
| Case 1, 09 April | 1.48$\pm$0.10 | 0.04$\pm$0.02 |
| Case 2, 24 April | 3.63$\pm$1.28 | -0.01$\pm$0.03 |
| Case 3, 15 April | 0.63$\pm$0.03 | 0.10$\pm$0.02 |
| Case 4, 03 April | 0.49$\pm$0.16 | 0.11$\pm$0.03 |

**Table 2.** A summary of optical properties derived from lidar observations in the case studies. The values before the '$\pm$' symbol represent the mean in the range of the layer height. The values after the '$\pm$' symbol represent the statistical error of the values before the symbol.

|  | Case 1: dust haze 15:00–16:00 09 April | Case 2: dust storm 15:00–16:00 24 April | Case 3: polluted dust 15:00–16:00 15 April | | Case 4: polluted dust 15:00–16:00 03 April | |
|---|---|---|---|---|---|---|
| Layer height [m] | 800–3800 | 800–2000 | 1000–2200 | 2400–2800 | 1000–1500 | 1500–3000 |
| $PLDR_{355}$ | 0.32$\pm$0.05 | 0.32$\pm$0.05 | 0.32$\pm$0.07 | 0.23$\pm$0.06 | 0.30–0.32$\pm$0.07 | 0.21$\pm$0.05–0.29$\pm$0.07 |
| $PLDR_{532}$ | 0.36$\pm$0.05 | 0.36$\pm$0.05 | 0.34$\pm$0.05 | 0.26$\pm$0.03 | 0.35$\pm$0.05 | 0.28$\pm$0.04–0.34$\pm$0.05 |
| $(V)PLDR_{1064}$ | 0.31$\pm$0.04[a] | – | 0.31$\pm$0.04 | 0.24$\pm$0.03 | 0.32$\pm$0.05 | 0.28$\pm$0.04–0.33$\pm$0.05 |
| $LR_{355}$ [sr] | 56$\pm$8 | 55[b] | 51$\pm$8 | 42$\pm$6 | 51$\pm$8 | 43$\pm$6–57$\pm$8 |
| $LR_{532}$ [sr] | 46$\pm$7 | 45[b] | 45$\pm$7 | 40$\pm$6 | 45$\pm$7 | 38$\pm$6–49$\pm$8 |
| $EAE_{355-532}$ | -0.01$\pm$0.30 | 0.01$\pm$0.30 | 0.02$\pm$0.30 | 0.10$\pm$0.30 | 0.02$\pm$0.30 | 0.14$\pm$0.30–0.30$\pm$0.30 |
| $BAE_{355-532}$ | -0.51$\pm$0.30 | – | -0.29$\pm$0.30 | -0.06$\pm$0.30 | -0.29$\pm$0.30 | -0.13$\pm$0.30–0.20$\pm$0.30 |
| RH [%] | 20$\pm$4–60$\pm$12 | 10$\pm$2–20$\pm$4 | 30$\pm$6–60$\pm$12 | 80$\pm$16 | 20$\pm$4–60$\pm$12 | 45$\pm$9–70$\pm$14 |
| WVMR [g/kg] | 2.2$\pm$0.5 | – | 3.5$\pm$7 | 4.0$\pm$0.8 | 2.7$\pm$0.6 | 2.7$\pm$0.6 |

[a] $PLDR_{1064}$ is not available in this case, but $VLDR_{1064}$ is. We assume $VLDR_{1064} \approx PLDR_{1064}$ considering aerosol scattering is much stronger than molecular scattering. [b] 55 and 45 sr are assumed lidar ratios based on the results in Case 1.

**Table 3.** A review of dust lidar ratios and particle linear depolarization ratios in literatures. The values of lidar ratios and PLDRs, as well as their errors are based on the results in the references. Error estimates are not provided if their are not available in the original publication.

| Dust source | Observation site | PLDRs | | | LRs | | Reference |
|---|---|---|---|---|---|---|---|
| | | 355 | 532 | 1064 | 355 | 532 | |
| Saharan dust | Ouarzazate[1a] | – | 0.30 | – | – | 38–50 | Esselborn et al. (2009) |
| | Ouarzazate[1b] | – | – | – | 53–55 | 53–55 | Tesche et al. (2009) |
| | Cape Verde | 0.24–0.27 | 0.29–0.31 | – | 48–70 | 48–70 | Groß et al. (2011) |
| | M'Bour[2a] | – | 0.34±0.05 | – | 68±10 | 50±8 | Veselovskii et al. (2016) |
| | M'Bour[2b] | – | 0.32±0.05 | – | 55–60±9 | 55–60±8 | Veselovskii et al. (2020) |
| | Leipzig | – | 0.15-0.25 | | 50–90 | 40–80 | Ansmann et al. (2003) |
| | Barbados | 0.26±0.03 | 0.27±0.01 | – | 53±5 | 56±7 | Groß et al. (2015) |
| | Barbados | 0.25±0.03 | 0.28±0.02 | 0.23±0.02 | 40–60 | 40–60 | Haarig et al. (2017) |
| Asian dust | Aksu | – | – | – | – | 42±3 | Jin et al. (2010) |
| | Japan | – | 0.20 | – | 49 | 43 | Murayama et al. (2004) |
| | Kazan | 0.23±0.02 | – | – | 43±14 | – | Dieudonné et al. (2015) |
| | Omsk | 0.17±0.02 | – | – | 50±13 | – | |
| | Dushanbe[3a] | 0.23±0.01 | 0.35±0.01 | – | 47±2 | 43±3 | Hofer et al. (2017) |
| | Dushanbe[3b] | 0.29±0.01 | 0.35±0.01 | – | 40±1 | 39±1 | |
| | Dushanbe[3c] | 0.24±0.03 | 0.33±0.04 | – | 43±3 | 39±4 | Hofer et al. (2020) |
| | UAE | 0.25±0.02 | 0.31±0.02 | – | 45±5 | 42±5 | Filioglou et al. (2020) |
| | Kashi | 0.28±0.07 – 0.32±0.07 | 0.36±0.05 | 0.31±0.05 | 51±8 – 56±8 | 45±7 | This study |
| American dust | Chihuahuan | 0.24±0.05 | 0.37±0.02 | 0.38±0.01 | – | – | Burton et al. (2015) |
| | Pico de Orizaba | – | 0.33±0.02 | 0.40±0.01 | – | – | Burton et al. (2015) |

[1a] HSRL measurements; [1b] Raman lidar measurement; [2a] 29 March 2015 in the dry season; [2b] 23–24 April 2015 in the transition period; [3a] Extreme dust case on 8 August 2015, [3b] Most extreme dust case on 14 July 2016; [3c] Statistical results estimated from 17 dust cases with $PLDR_{532} > 0.31$.

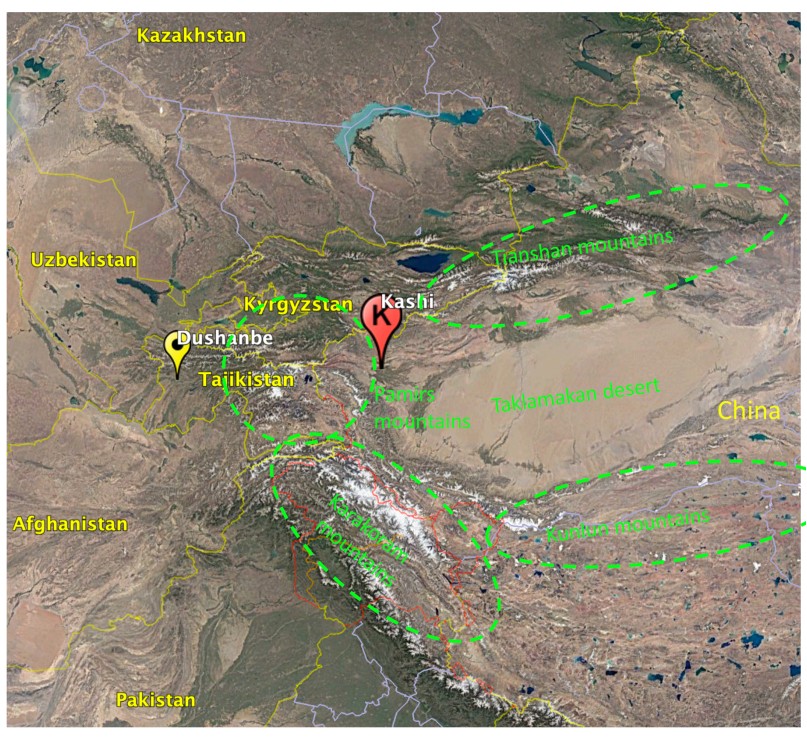

**Figure 1.** The location of the observation site in Kashi (at 39.51N, 75.93E). The observation site is about 628 km in the east of Dushanbe, Tajikistan (38.53N, 68.77E ). The green ellipses indicate the mountain ranges surrounding the Taklamakan desert, including the Tianshan mountains, the Pamir mountains, the Karakoram mountains, the Kunlun and Altun mountains. @ Google Maps 2020.

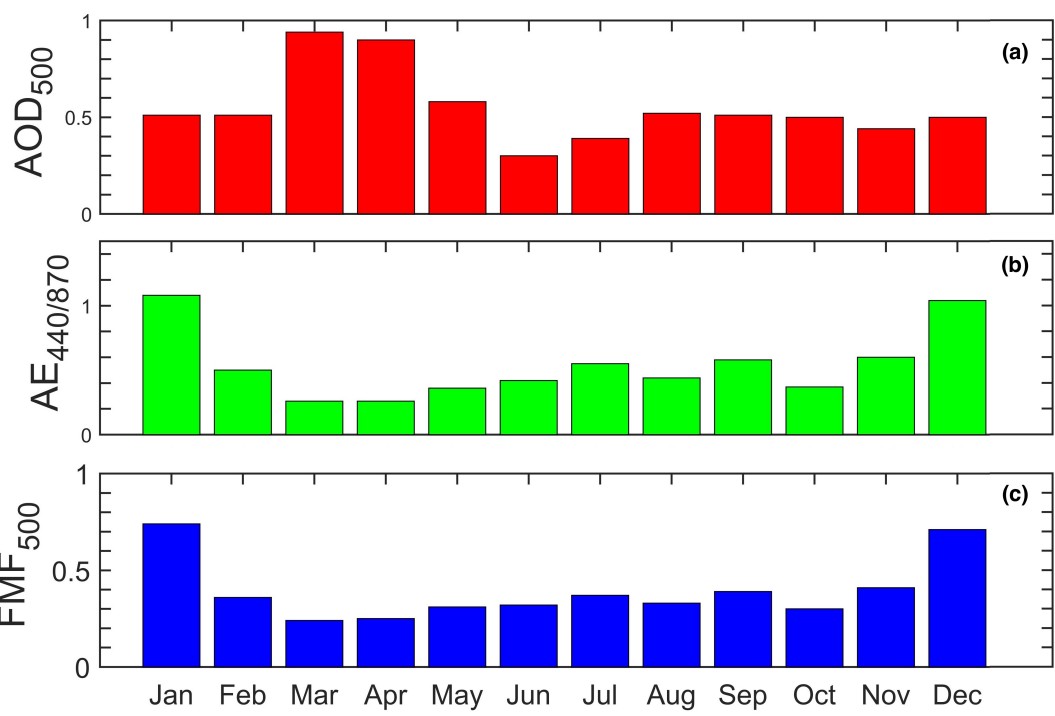

**Figure 2.** Monthly means of (a) the AOD at 500 nm, (b) Angström exponent (440–870) and (c) FMF at 500 nm from 2013 to 2017. The data are obtained from the SONET network.

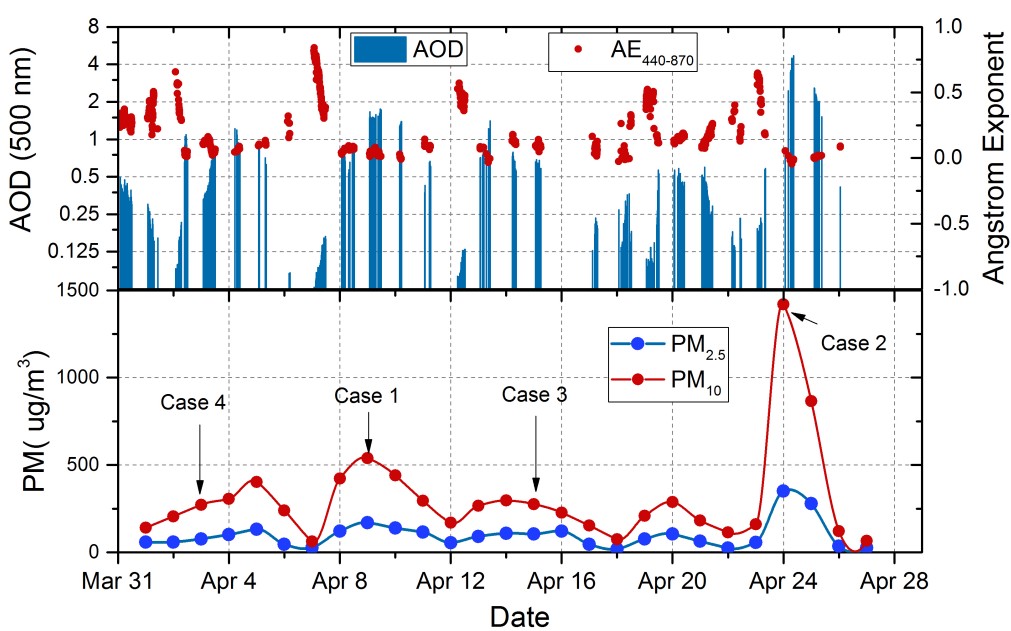

**Figure 3.** The AOD at 500 nm, Angström exponent (440–870) and daily particulate matter (in $\mu g\,m^{-3}$) in April 2019. The AODs are measured by the sun/sky photometer deployed in Kashi site, and the data are stored in the SONET network. The particulate matter measurements are public data from a meteorological station, 5 km to the observation site.

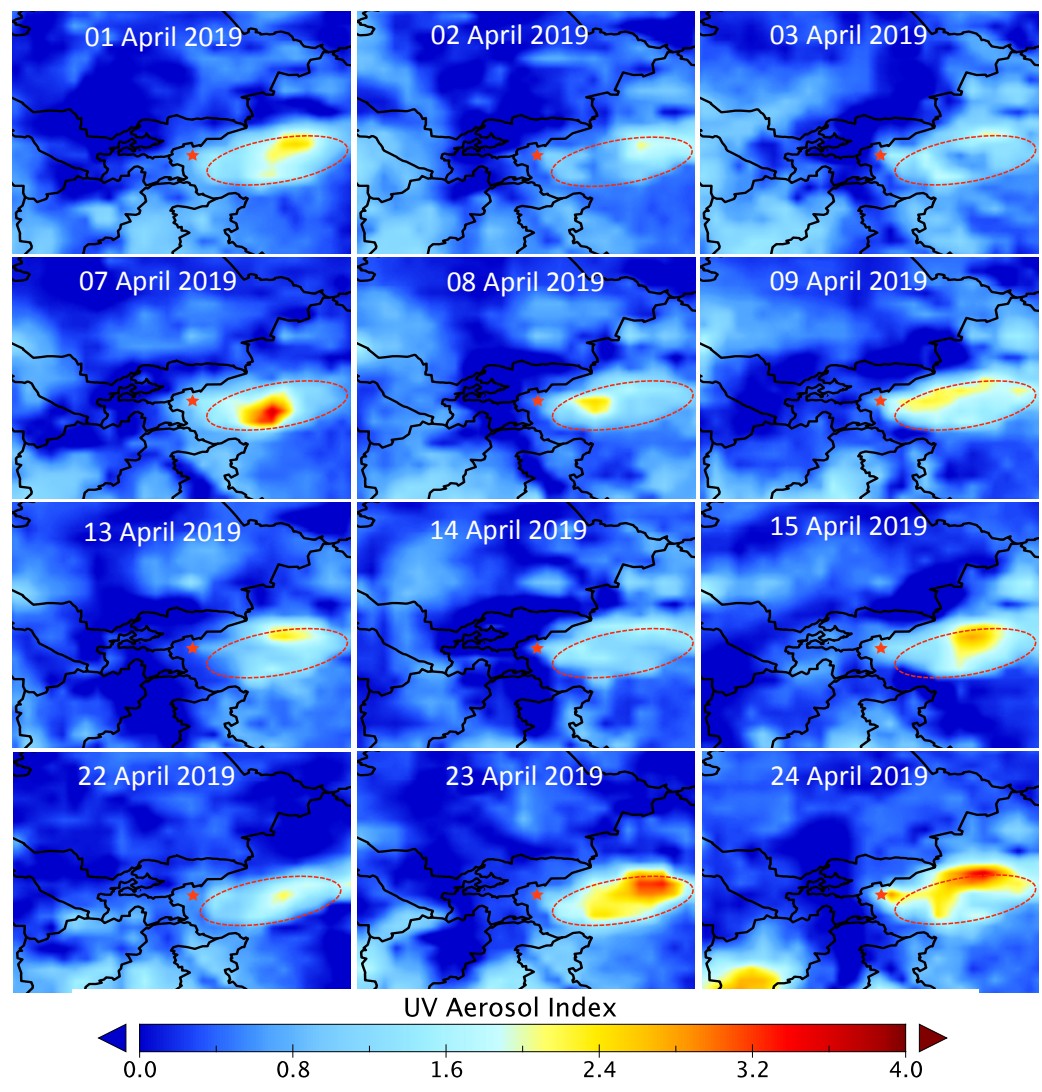

**Figure 4.** The UVAI derived from OMPS instrument onboard the Suomi-NPP satellite. The red star represents the location of the observation site. The dashed red ellipse represents the location of the Taklamakan desert.

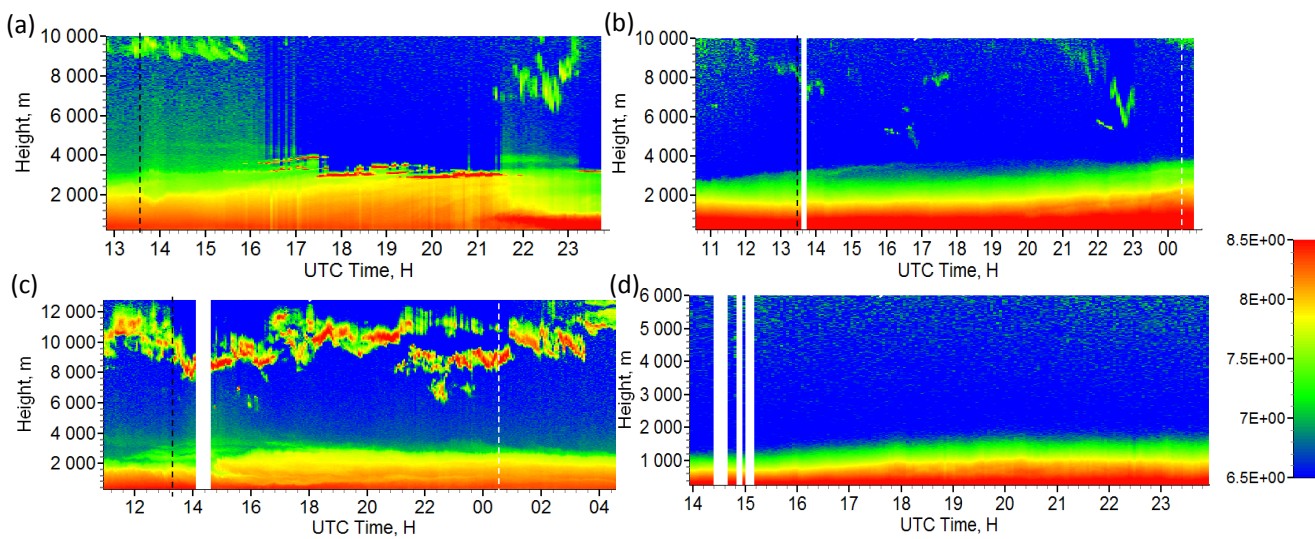

**Figure 5.** The quicklooks of the range-corrected lidar signal at 532 nm in the four cases: (a) Case 4: 03 April 2019, (b) Case 1: 09 April 2019, (c) Case 3: 15 April 2019and (d) Case 2: 24 April 2019. The dashed black lines represent the sunset time and dashed white lines represent the sunrise time.

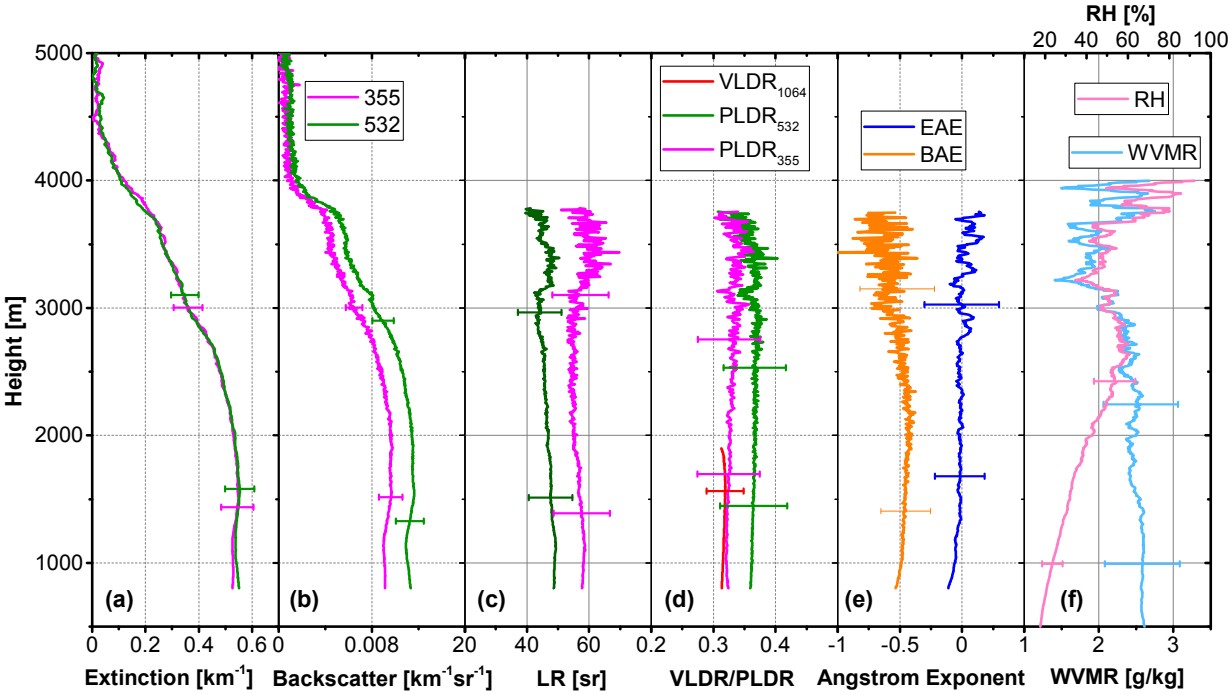

**Figure 6.** Case 1: Lidar derived parameters at 17:00–22:00 UTC, 09 April 2019. (a) Extinction coefficient, (b) backscattering coefficient, (c) lidar ratio, (d) PLDR/VLDR, (e) $EAE_{355-532}$ and $BAE_{355-532}$, (f) WVMR and RH.

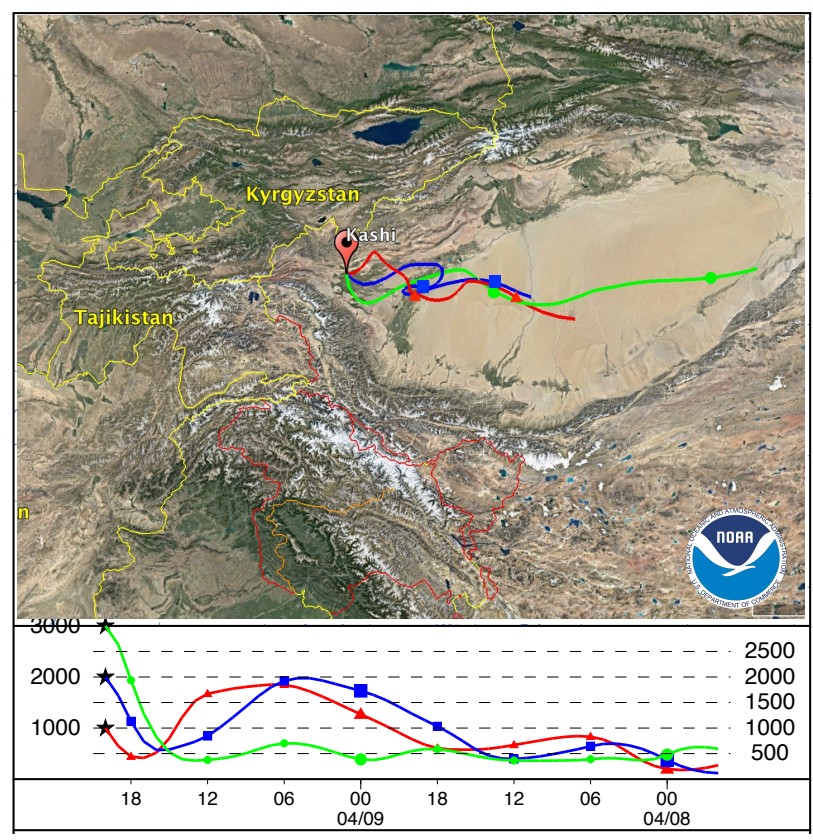

**Figure 7.** Case 1: The 48-hour back trajectories ending at 20:00 UTC, 09 April 2019 for air mass at 1000, 2000 and 3000 m. @ Google Maps 2020.

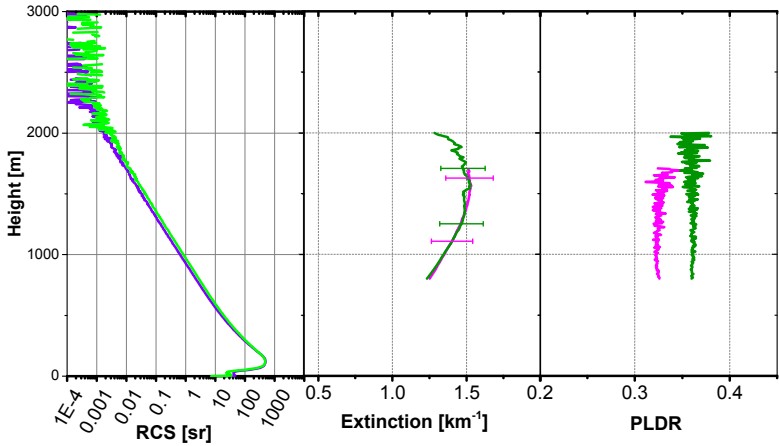

**Figure 8.** Case 2: Lidar derived parameters at 15:00–24:00 UTC, 24 April 2019. (a) The Raman lidar signals at 530 and 387 nm. (b) The extinction coefficients at 355 and 532 nm. (c) The PLDRs at 355 and 532 nm.

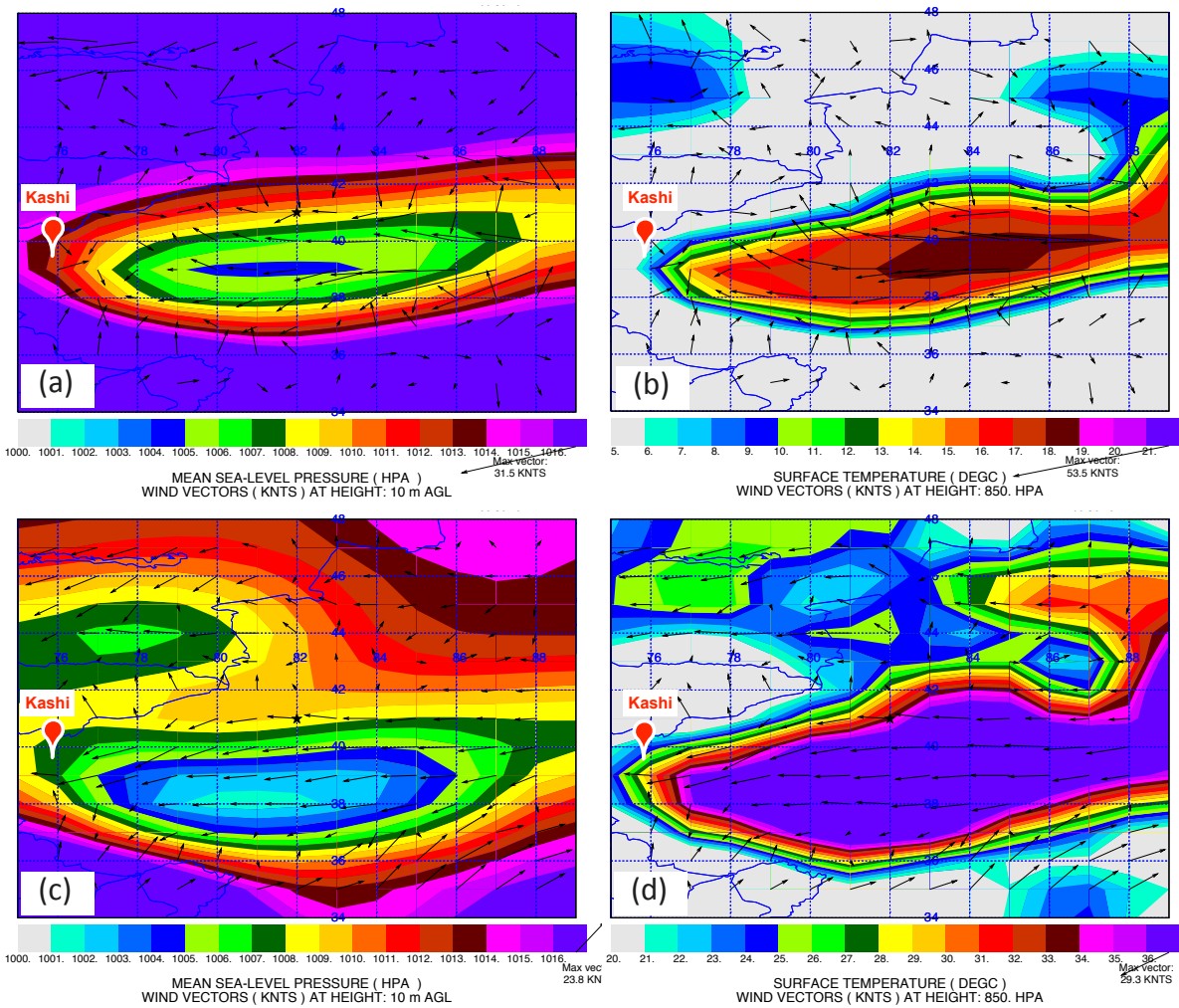

**Figure 9.** Case 2: The synoptic condition at 00:00 UTC, 23 April (a, b) and 06:00 UTC, 24 April (c, d), 2019. The data are obtained from the 1-degree GDAS archived meteorological data. (a) and (c) The mean sea level pressure at the surface overlaid with wind vector at 10 m above the ground level. (b) and (d) The temperature at 2 m vertical level overlaid with wind vector at 850 hPa vertical level.

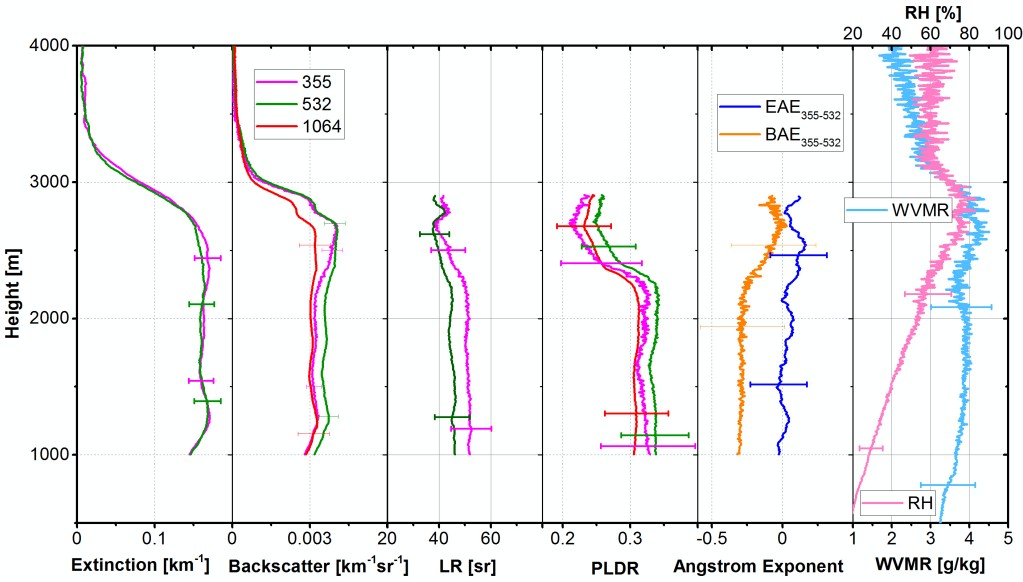

**Figure 10.** Case 3: Lidar derived parameters at 18:00–20:00 UTC, 15 April 2019. (a) Extinction coefficient, (b) backscattering coefficient, (c) lidar ratio, (d) PLDR, (e) $EAE_{355-532}$ and $BAE_{355-532}$, (f) WVMR and RH.

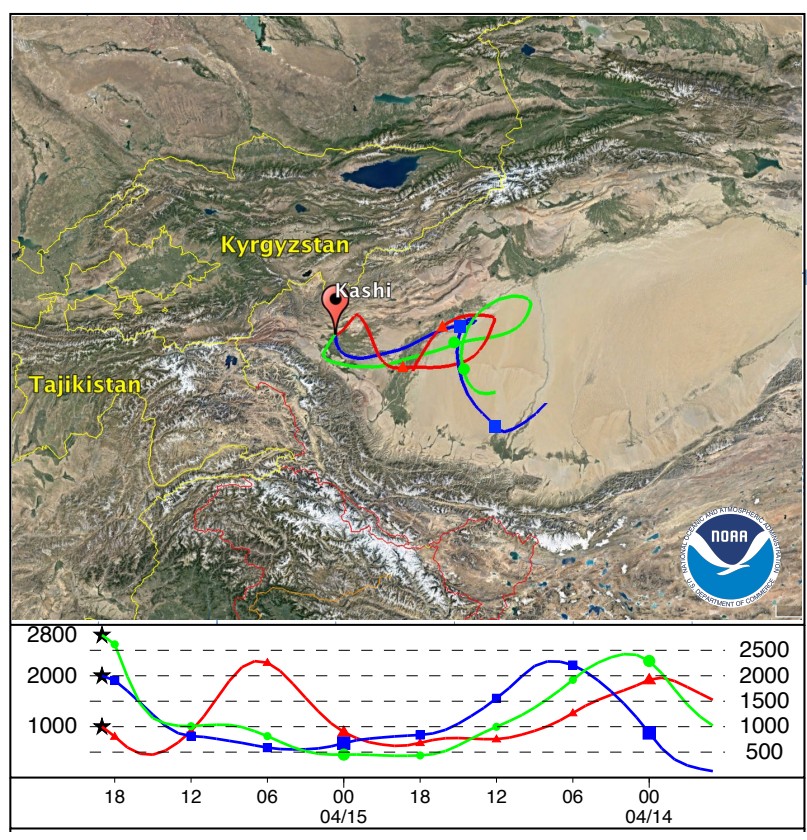

**Figure 11.** Case 3: The 48-hour back trajectories ending at 19:00 UTC, 15 April 2019 for air mass at 1000, 2000 and 2800 m. @ Google Maps 2020.

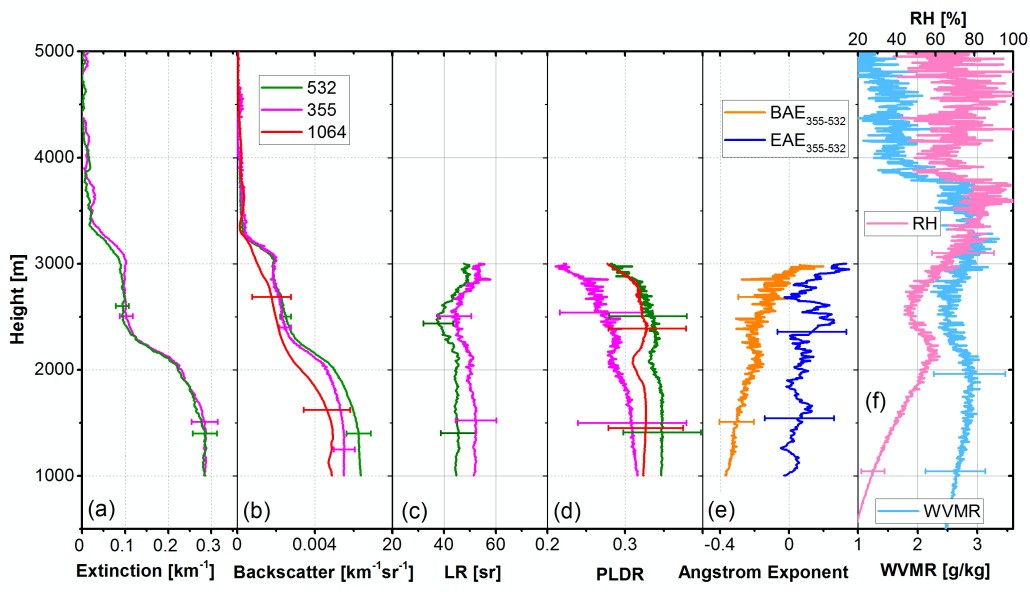

**Figure 12.** Case 4: Lidar derived parameters at 14:00–16:00 UTC, 03 April 2019. (a) Extinction coefficient, (b) backscattering coefficient, (c) lidar ratio, (d) PLDR, (e) $EAE_{355-532}$ and $BAE_{355-532}$, (f) WVMR and RH.

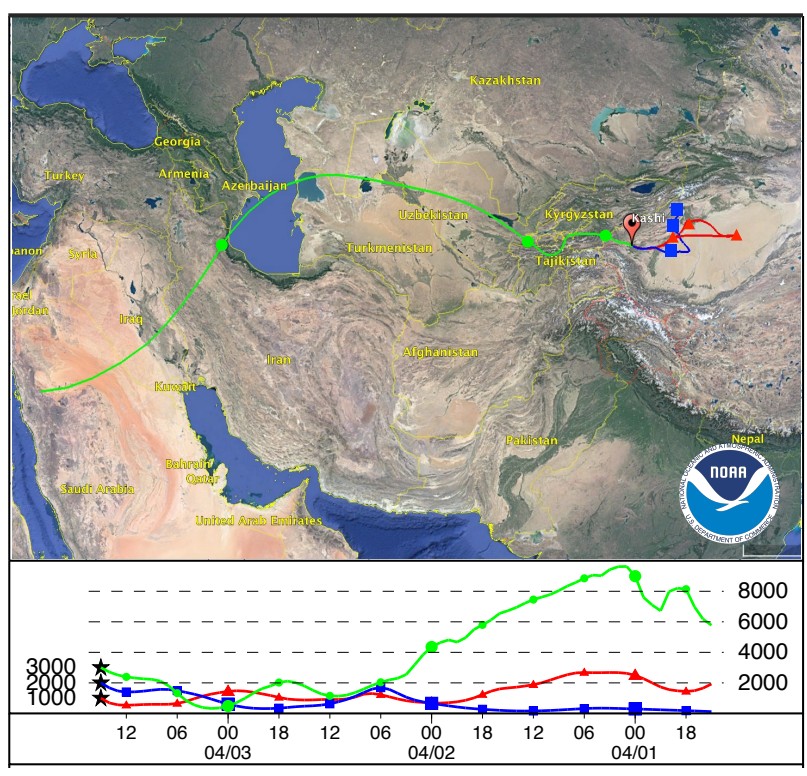

**Figure 13.** Case 4: The 72-hour back trajectories ending at 15:00 UTC, 03 April 2019 for air mass at 1000, 2000 and 3000 m. @ Google Maps 2020.

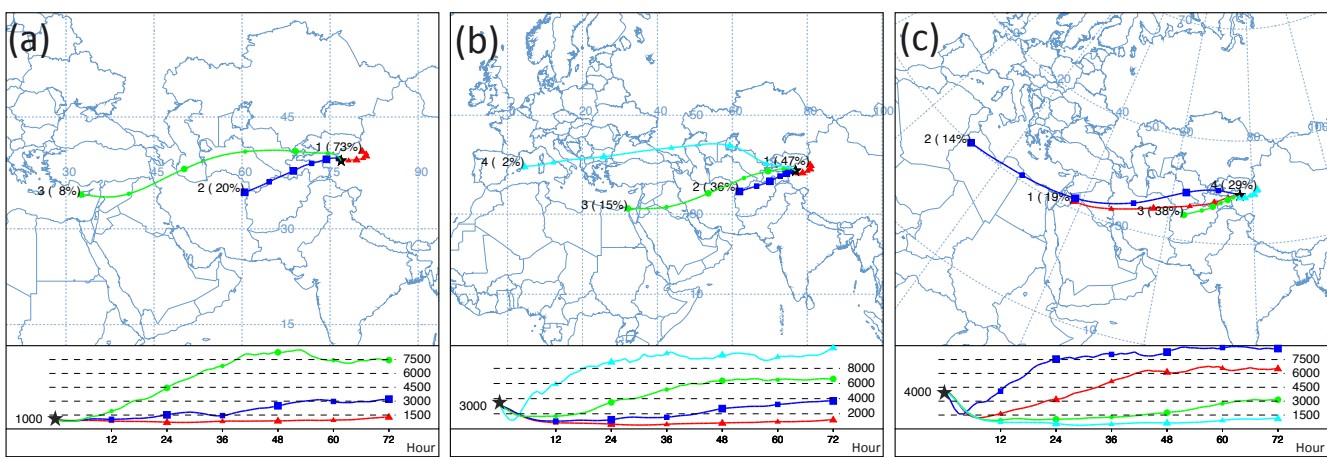

**Figure 14.** The clustering of air mass in April 2019. The clustering is performed using HYSPLIT and based on back trajectories with a 2-hour time resolution and 72-hour duration. (a) 1000 m, (b) 3000 m, (c) 4000 m.