# Peer review of "The characterization of Taklamakan dust properties using a multi-wavelength Raman polarization lidar in Kashi, China"

_Atmospheric Chemistry and Physics, 2020_

## Referee Comment (RC1) · Anonymous Referee #2 · 19 Jun 2020

**General comments**

This paper reports the lidar observations in Kashi, China located west of the Taklamakan Desert. The location is very interesting, and the quality of the observations with a multi-wavelength Raman lidar looks high, and consequently the results merit publication. However, the discussion on dust characterization in the present manuscript is only conceptual and very ambiguous. No strong conclusions are obtained. The authors discuss polluted dust cases, but the definition of polluted dust is not clear. The location of the observation is relatively clean except for desert dust. Is the polluted dust a mixture with anthropogenic air pollution? Is it external mixture or internal mixture?

[Figure]

Probably, it would not be possible to characterize it only with lidar data. Variability of the characteristics of "pure" dust is not well understood. Also, optical parameters are dependent on particle size distribution even if the composition is the same.

The manuscript should be rewritten, in my opinion, with more focus on detailed comparison of the observed parameters (lidar ratio, particle depolarization ratio, Angstrom exponents) with previously reported results. The discussion with Table 3 in the present manuscript is not sufficient. Discussion on the change in optical characteristics by mixing with pollution should be given, if "polluted dust" is discussed.

Detailed comments

Line 19: T yr-1 -> Tg yr-1

Line 28-35: The authors should describe how the lidar data can be used as input and validation of models.

Line 75-77 "Moreover, there are populated cities in the neighboring countries such as Kyrgyzstan, Tajikistan and Pakistan. Under favorable meteorological conditions, various aerosol, for example, pollution, could be potentially transported to Kashi and mix with dust aerosols. ": This statement is not convincing, looking at the map.

Line 96-98: To my knowledge, the error analysis cannot be this simple. The error in extinction must be different from that in backscatter. Also, the error must be dependent on height and the background radiation. It should be mentioned that the Raman lidar measurement was limited in the nighttime, if so. In addition, it would be better to have some descriptions about the advantage of using rotational Raman instead of vibrational Raman at 532 nm.

Figure 3: The periods of Case1, 2, 3 and 4 should be indicated in Figure 3.

Figure 5: Case1, 2, 3 and 4 should be indicated in Figure 5.

Figure 3: Legend "500 nm" should be AOD (500 nm).

[Figure]

Line 166-168: The backscatter coefficient at 1064 nm below 1800 m should be indicated in Fig. 6.

Line 169-170: "EAE" and "BAE" are not defined.

Line 183-187: Is the description consistent with Figure 3?

Line 227-228: What is the "clear evidence of polluted dust"?

Line 229-232: The structure at around 2500 km is interesting and should be studied further. Is the type of dust (or "polluted" dust) the same in 1000-2200 m and 2400-2800 m or different? Why relative humidity was high in 2400-2800 m?

Figure 9: Captions for (c) and (d) are missing. The scales (color scale and vector) should be the same for all panels.

---

## Referee Comment (RC2) · Anonymous Referee #1 · 23 Jul 2020

A unique data set of mineral dust optical properties taken in western China close to the Taklamakan desert is presented. The observations were performed with an advanced multiwavelength Raman and polarization lidar. It is probably (almost) impossible for non-Chinese research teams to travel to the westernmost part of China. This makes the data set so valuable.

The paper is well written. I recommend publication after minor revision.

Abstract: Do we need all these details to all 4 discussed cases in the abstract? A few summarizing sentences would be sufficient to my opinion!

Introduction: P2,L26: Please mention the important role of dust particles to serve

as ice-nucleating particles, reference: Kanji, Z. A., Ladino, L. A., Wex, H., Boose, Y., Burkert-Kohn, M., Cziczo, D. J., and Kramer, M.: Chapter 1: Overview of ice nucleating particles, Meteor Monogr., Am. Meteorol. Soc., 58, 1.1-1.33, https://doi.org/10.1175/amsmonographs-d-16-0006.1, 2017.

P2, L46: Besides the Hofer 2017 paper there are two additional Hofer papers in 2020. The last of these three articles is on lidar ratios and depolarization ratios measured '. . .a few kilometers upwind. . .' of your Kashi lidar station. This paper should be used for comparison regarding the potential impact of long-range transport of dust and pollution advected from Africa and the Middle East.

P4, L92: Basic information about the methods (Fernald, Raman, smoothing lengths, least squares fit, reference height in backscatter determination, input parameters) regarding the computation of the backscatter and extinction coefficients would be fine. The same for the retrieval of the particle depolarization ratio from the volume depolarization ratios. Information on the used temperature and pressure profiles is required. Did you use GDAS profiles? Kashi is a radiosonde station, that means the re-analyzed GDAS data consider these radiosonde observations and are thus perfect to be used in your lidar data analysis.

P4, L116: Improve I. . . , better write I340 and I380 represent . . .

P5, L142: Are you sure that the photometer can correctly measure an AOD of 4.7?

P6, Case 1: The depolarization ratios point to pure dust, and more important, to near-source dust with a large fraction of coarse particles and especially giant particles (radius > 20 microns). This is probably the reason for the strong difference between the lidar ratio at 355 nm of around 60 sr and of 45 sr for 532 nm and the corresponding backscatter wavelength dependence. The Dushanbe observations (Hofer papers) of central Asian, Saharan, and Middle East dust did not show that. Should be discussed.

P6-7: Case 2 is almost 'no case', and indicates again the dominance of giant dust

particles, causing these extremely large particle depolarization ratios of 0.32 at 355 nm and 0.37 at 532 nm. It should be mentioned that the depolarization ratios were exceptionally high because of the presence of very large particles. Burton et al. (ACP) measured very high depolarization ratios at 532 nm close to dust sources, but never at 355 nm. Should be discussed.

P7-8: Case 3: You mention that this is a polluted case, ..and dust was contaminated and coated. Do you have clear indications for that? There is long debate on external or internal mixture of dust and pollution aerosol. Researchers (e.g., Kandler and his team) who investigated Saharan dust particles in the Caribbean did not find any significant coating. They found the same during the SAMUM-2 campaign with strong pollution and dust mixtures. Kandler did not find strong hinds on coating and concluded that dust and pollution is mainly externally mixed.

If you do not have clear hinds on coating then one should clearly indicate that by writing…. we hypothesize that dust is coated or so…..

P8-9: Case 4: This dust case is ideal to compare all the numbers with the findings of Hofer et al. (2020) on lidar and depol ratios.

Discussion: Again, please state clearly that the measurements are taken at a site rather close to a strong dust source so that giant particles have a strong impact on the measurements. This is not the case for almost all the observations published in the literature. After 1000 km travel most giant particles are gone, and the influence of fine dust on the optical properties increases. There is always fine-mode dust and coarse mode dust and giant-mode dust. Fine dust produces depolarization ratio below 20% at 532 and 1064 nm. Not only pollution aerosol can lead to a decrease of the depolarization ratio.

P10, P286: I am a bit surprised that you did not mentioned the Hofer et al. papers in this context! Should be improved.

It is good to have Table 3 for comparison and discussion. Please check Hofer 2020 (on lidar ratios and depol ratios) and include it here.

Figure 1: Kashi is at 39.47N and 75.98E, is the lidar field site really at 74.95 E as indicated in Figure 1? By the way, you could even include Dushanbe at 38.53N and 68.77 E in the map.

Figure 3: PM10 does not include the contribution by giant particles. Visibility observations (at the Kashi airport?) would be nice and conversion of the visibility-related extinction coefficients into mass concentrations... That would then clearly show the impact of giant particles.

Figure 6: (a) the height profiles of the extinction coefficients are fine and indicates large particles. But why is the 532 nm backscatter coefficient always larger than the 355 nm backscatter coefficient, even above the dust layer at heights above 4 km? I would assume that giant particles are not present anymore at such large heights, and clearly above the main dust layer. Please check the data analysis.

Figure 11 indicates similar air mass flow at all heights from 1000 to 3000 m.

Figure 10: According to Fig.11 the extinction profiles and the 532 and 1064 nm backscatter and depolarization ratio profiles are fine. But I have always a bit my doubts concerning the 355 nm backscatter and depolarization ratio values. If the particle backscatter profile is a bit wrong in the case of 355 nm then the particle depol. ratio will be wrong as well. The conversion from volume to particle depol ratio is very sensitive to the 355 nm backscatter values.

---

## Author Comment (AC1) · 3 Sep 2020

Thanks to the reviewer for the very helpful advice. We appreciate the reviewer's help and effort in reviewing this paper. The answers to the reviwers' are listed below.

A unique data set of mineral dust optical properties taken in western China close to the Taklamakan desert is presented. The observations were performed with an advanced multiwavelength Raman and polarization lidar. It is probably (almost) impossible for non-Chinese research teams to travel to the westernmost part of China. This makes the data set so valuable.

The paper is well written. I recommend publication after minor revision.

Abstract: Do we need all these details to all 4 discussed cases in the abstract? A few summarizing sentences would be sufficient to my opinion!

Answer: The abstract has been modified.

Introduction: P2,L26: Please mention the important role of dust particles to serve as ice-nucleating particles, reference: Kanji, Z. A., Ladino, L. A., Wex, H., Boose, Y., Burkert-Kohn, M., Cziczo, D. J., and Kramer, M.: Chapter 1: Overview of ice nucleating particles, Meteor Monogr., Am. Meteorol. Soc., 58, 1.1-1.33, https://doi.org/10.1175/amsmonographs-d-16-0006.1, 2017.

Answer: It is added. Thanks for the suggestion.

P2, L46: Besides the Hofer 2017 paper there are two additional Hofer papers in 2020. The last of these three articles is on lidar ratios and depolarization ratios measured '…a few kilometers upwind…' of your Kashi lidar station. This paper should be used for comparison regarding the potential impact of long-range transport of dust and pollution advected from Africa and the Middle East.

Answer: Hofer et al. 2020 has been cited and used as comparison in the manuscript.

P4, L92: Basic information about the methods (Fernald, Raman, smoothing lengths, least squares fit, reference height in backscatter determination, input parameters) regarding the computation of the backscatter and extinction coefficients would be fine. The same for the retrieval of the particle depolarization ratio from the volume depolarization ratios. Information on the used temperature and pressure profiles is required. Did you use GDAS profiles? Kashi is a radiosonde station, that means the re-analyzed GDAS data consider these radiosonde observations and are thus perfect to be used in your lidar data analysis.

Answer: A brief introduction to the input parameters in the calculation has been added in the manuscript. We did not use the re-analyzed data from GDAS. The temperature and pressure profiles needed in the data processing are from the radio sounding measurements obtained from the radiosonde site ~6 km to the observation site. There are 2 measurements per day. The time difference between our lidar measurements and the radiosonde measurements is about a few hours.

P4, L116: Improve I… , better write I340 and I380 represent…

Answer: It has been corrected in the manuscript.

P5, L142: Are you sure that the photometer can correctly measure an AOD of 4.7?

Answer: It is true that the AOD =4.7 is touching the limit of the capability of the sun photometer. The measurement was taken by an old version sun photometer who max AOD could be about 4.0 (With the new version photometer, CE318-N, the max AOD could reach 7.0). Under this condition, the incident solar radiation was very weak, but not that weak because the solar zenith angle is still enough. So the accuracy of the the detection might decrease but not the result should not be rediculously wrong. A brief explanation been added in the manuscript: "It should be noted, in this extreme case, the accuracy of the measured AOD (i.e. 4.70) may degrade because of decreased signal-to-noise ratio. " By the way, we are using this value to prove, qualitatively, that the AOD was extremely high, and never used it in any scientific calculation.

P6, Case 1: The depolarization ratios point to pure dust, and more important, to nearsource dust with a large fraction of coarse particles and especially giant particles (radius > 20 microns). This is probably the reason for the strong difference between the lidar ratio at 355 nm of around 60 sr and of 45 sr for 532 nm and the corresponding backscatter wavelength dependence. The Dushanbe observations (Hofer papers) of central Asian, Saharan, and Middle East dust did not show that. Should be discussed.
Answer: Thank you for the advice. This argument has been added in the presentation of Case 1 and in the discussion part.

P6-7: Case 2 is almost 'no case', and indicates again the dominance of giant dust particles, causing these extremely large particle depolarization ratios of 0.32 at 355 nm and 0.37 at 532 nm. It should be mentioned that the depolarization ratios were exceptionally high because of the presence of very large particles. Burton et al. (ACP) measured very high depolarization ratios at 532 nm close to dust sources, but never at 355 nm. Should be discussed.
Answer: A short discussion has been added in the end of Case 2.

P7-8: Case 3: You mention that this is a polluted case, ..and dust was contaminated and coated. Do you have clear indications for that? There is long debate on external or internal mixture of dust and pollution aerosol. Researchers (e.g., Kandler and his team) who investigated Saharan dust particles in the Caribbean did not find any significant coating. They found the same during the SAMUM-2 campaign with strong pollution and dust mixtures. Kandler did not find strong hinds on coating and concluded that dust and pollution is mainly externally mixed. If you do not have clear hinds on coating then one should clearly indicate that by writing… we hypothesize that dust is coated or so….
Answer: We realized that it is not cautious to say "…but when *coated* by hygroscopic aerosol species…". Due to the lack of aerosol samples at the boundary layer top, we do not have enough evidence to tell the occurrence of hygroscopic growth and the mix00ing state of dust and pollution. The above mentionned paragraph has been rewritten:
 *"but when mixed with hygroscopic aerosol species, for example, nitrate, the ensemble of aerosol mixture could become hygroscopic. The fine mode particles can be hydrophobic or hydroscopic, depending on their chemical compositions. In this case, there were no no clear evidence indicating the occurrence of hygroscopic growth or the mixing state of dust and pollution particles."*

P8-9: Case 4: This dust case is ideal to compare all the numbers with the findings of Hofer et al. (2020) on lidar and depol ratios.

Answer: The comparison to Hofer et al. 2020 has been added. Some sentenses referring to Hofer et al. 2020 have been added in Case 4.

Discussion: Again, please state clearly that the measurements are taken at a site rather close to a strong dust source so that giant particles have a strong impact on the measurements. This is not the case for almost all the observations published in the literature. After 1000 km travel most giant particles are gone, and the influence of fine dust on the optical properties increases. There is always fine-mode dust and coarse mode dust and giant-mode dust. Fine dust produces depolarization ratio below 20% at 532 and 1064 nm. Not only pollution aerosol can lead to a decrease of the depolarization ratio.

Answer: Thank you for the suggestion. It is added in the manuscript.

P10, P286: I am a bit surprised that you did not mentioned the Hofer et al. papers in this context! Should be improved. It is good to have Table 3 for comparison and discussion. Please check Hofer 2020 (on lidar ratios and depol ratios) and include it here.

Answer: Hofer et al. 2020 (on lidar ratio and depolarization) has been included in Table 3.

Figure 1: Kashi is at 39.47N and 75.98E, is the lidar field site really at 74.95 E as indicated in Figure 1? By the way, you could even include Dushanbe at 38.53N and 68.77 E in the map.

Answer: As indicated in the manuscript, the observation site was located at 39.51N, 75.93E, which is in the northwest of the Kashi city. The orthogonal lines labeling longitude and latitude are not well aligned because the base map was in a 3D globe mode, not flat. To simplify the map (because there are already too many elements on the map), the author decided to remove the label of latitudes and longitudes. In addition, Dushanbe has been added on the map. Thanks to the reviewer's suggestion.

Figure 3: PM10 does not include the contribution by giant particles. Visibility observations (at the Kashi airport?) would be nice and conversion of the visibility-related extinction coefficients into mass concentrations…That would then clearly show the impact of giant particles.

Answer: we are agree that the PM10 data do not include giant particles with radius greater than 20 microns. The visibility data were for Kashi airport and data are public on the website: https://www.timeanddate.com/weather/china/kashgar/historic?month=4&year=2019 But we cannot assure the quality of the data and have no information about how these measurements were made. We referred the values of visibility to show that dust content was extremely high, but we do not use the visibility data for calculation.

Figure 6: (a) the height profiles of the extinction coefficients are fine and indicates large particles. But why is the 532 nm backscatter coefficient always larger than the 355 nm backscatter coefficient, even above the dust layer at heights above 4 km? I would assume that giant particles are not present anymore at such large heights, and clearly above the main dust layer. Please check the data analysis.

Answer: We have checked the data analysis, there is no sign showing a decrease versus height in the spectral dependency of backscattering coefficient between 355 and 532 nm. Moreover,

the vertical variability of the PLDRs is also not very important. It indicates that particles were mixed well. This case on 09 April started from the morning of 08 April, very strong convection injected dust from the surface to the boundary layer. This event settled down in the night of 10 April. In the 3-day observations, we did not see any significant vertical variations of the backscatter-related Angstrom exponent and the PLDRs in the dust layer.

Figure 11 indicates similar air mass flow at all heights from 1000 to 3000 m.

Answer: Indeed, the back trajectory indicates the air mass at the three different levels are all originated from the same region, which is the west of the Taklamakan desert. From the UVAI maps, we can see that there were no evident dust activities during the overpass of the air mass. This explains the relatively low dust content, observed by the lidar. While the aerosol properties shown by the lidar profiles present distinct characteristics, showing features of pure dust in the lower boundary layer and polluted dust in the upper boundary layer. If we look into the trajectories of air mass when they are approaching the observation site, we can see that the air mass in upper boundary layer were lifted from near the surface in the urban region, while air mass in lower boundary layers were descending from the rural region. So, the air mass in upper boundary layer are more possible to mixed with some anthropogenic components. Moreover, air mass clustering in Figure 14 shows that, statistically, a nonnegligible proportion air mass at upper levels is originated from the long-distance west-to-east transport. This process may not be captured by a single back trajectory.

Figure 10: According to Fig.11 the extinction profiles and the 532 and 1064 nm backscatter and depolarization ratio profiles are fine. But I have always a bit my doubts concerning the 355 nm backscatter and depolarization ratio values. If the particle backscatter profile is a bit wrong in the case of 355 nm then the particle depol. Ratio will be wrong as well. The conversion from volume to particle depol ratio is very sensitive to the 355 nm backscatter values.

Answer: We agree that the errors in the backscattering coefficient will directly enter into the particle linear depolarization ratio. The error of PLDR is estimated accounting for the error of the backscatter coefficient, the volume depolarization ratio and the molecular depolarization ratio (Hu et al. 2019). And the error for PLDR at 355 nm is about 15% for dust cases (assuming 10% of error in the backscattering coefficient profile). An example is give below. We were also surprised when we found so high PLDR at 355 nm. But during the one-month observation, we found this value is very stable. Although the aerosol content changed, the mean PLDR at 355 nm varied 0.29—0.32 in dust from the Taklamakan desert. In addition, simultaneous cloud observations (in the night of 15-16 April 2019) showed that the PLDR at 355 nm for clouds at 9500-11500 m was in the range of 0.38-0.45, which are reasonable values. Therefore, we think this high PLDR at 355 nm is realistic and is resulted from the coarse-mode and giant particles in fresh dust.

Case 1:

| Wavelength (nm) | R | VLDR | MDR | E_R | E_VLDR | E_MDR | PLDR | E_PLDR |
|---|---|---|---|---|---|---|---|---|
| 355 | 2.6 | 0.19 | 0.015 | 10% | 10% | 200% | 0.33 | 15% |
| 532 | 9.80 | 0.31 | 0.020 | 10% | 10% | 300% | 0.36 | 11% |

---

## Author Comment (AC2) · 3 Sep 2020

Thanks to the reviewer for his/her helpful advice. We appreciate the reviewer's help and effort in reviewing this paper. The answers to the reviwers' are listed below.

General comments

This paper reports the lidar observations in Kashi, China located west of the Taklamakan Desert. The location is very interesting, and the quality of the observations with a multi-wavelength Raman lidar looks high, and consequently the results merit publication. However, the discussion on dust characterization in the present manuscript is only conceptual and very ambiguous. No strong conclusions are obtained. The authors discuss polluted dust cases, but the definition of polluted dust is not clear. The location of the observation is relatively clean except for desert dust. Is the polluted dust a mixture with anthropogenic air pollution? Is it external mixture or internal mixture? Probably, it would not be possible to characterize it only with lidar data.

Answer:  We have to admit that, with lidar measurements, we are not able to provide further information to tell the exact species of the pollutants nor the mixing state of dust and pollution, although they are very important information. To obtain such information and to get "strong conclusion", in-situ measurements are required.

According to long-term observations in <Li et al. 2018: Comprehensive study of optical, physical, chemical, and radiative properties of total columnar atmospheric aerosols over China: an overview of sun--sky radiometer observation network (SONET) measurements> and Figure 2, the role of anthropogenic aerosol is not negligible, so Kashi cannot be simply regarded as a 'clean site'.  Kashi is a populated city in Xinjiang (see the figure below, referring to <Doxsey-Whitfield, Erin, et al. "Taking advantage of the improved availability of census data: a first look at the gridded population of the world, version 4." Papers in Applied Geography 1.3 (2015): 226-234.>), anthropogenic emission should be reasonably expected to occur. As to the mixing state, it is out of the scope of this paper and beyond what we can obtain on the basis of what we have.

[Figure]

Variability of the characteristics of "pure" dust is not well understood. Also, optical parameters are dependent on particle size distribution even if the composition is the same. The manuscript should be rewritten, in my opinion, with more focus on detailed comparison of the observed parameters (lidar ratio, particle depolarization ratio, Angstrom exponents) with previously reported results. The discussion with Table 3 in the present manuscript is not

sufficient. Discussion on the change in optical characteristics by mixing with pollution should be given, if "polluted dust" is discussed.

Answer: The definition of 'pure dust' is given in the beginning of the 'Discussion' section. In this paper, the 'pure' Taklamakan dust is defined with PLDR >0.32 at 532 nm and the EAE(355-532) smaller than 0.1. The identification of Taklamakan dust is also confirmed with back trajectory. The definition of polluted dust is "PLDR <0.3 at 532 nm and EAE >0.2". Again, back trajectory is used to support the identification. We agree that the optical properties are dependent on not only the composition but also the size distribution. For example, dust aerosol with different fraction of fine dust could present different optical properties, such as BAE, PLDR… This issue is added in the manuscript and the discussion part is improved.

Detailed comments
Line 19: T yr-1 -> Tg yr-1
Answer: Corrected.

Line 28-35: The authors should describe how the lidar data can be used as input and validation of models.

Answer: A common way of involving lidar data into models is to simulate lidar profiles (of lidar signal, backscatter coefficient profile, extinction profile or depolarization profile) with the output or description of models for a model-given atmospheric state. For example, Sekiyama et al. 2010 assimilated the backscatter coefficient and depolarization profiles of CALIPSO Level 1B data. In the model, the backscatter coefficient equals to the sum of backscatter coefficients of several aerosol component, such as sulfate, sea-salt and dust, whose concentrations are model prognostic variables. Zhang et al. 2011 and Campbell et al. 2010 chose to deal with the extinction coefficient of CALIPSO in mass transport model. Apart from satellite lidar, modelers also used ground-based lidar measurements as input of models. Wang et al. 2013 used AirBase lidar network data to simulate PM10 concentrations. As to model validation, it mostly depends on the output of models and the variables to be validated. In Yu et al. 2010, both vertical profiles, e.x. extinction profile, and integrated variable, e.x. AOD from CALIPSO are used to validate the GOCART model. However, the authors consider this detailed information is not so relevant to the topic of our paper. So, a brief description and a list of references given in the manuscript will be sufficient.

Line 75-77 "Moreover, there are populated cities in the neighboring countries such as Kyrgyzstan, Tajikistan and Pakistan. Under favorable meteorological conditions, various aerosol, for example, pollution, could be potentially transported to Kashi and mix with dust aerosols. ": This statement is not convincing, looking at the map.

Answer: In Figure 14, the air mass clustering indicates that the air mass arriving at the observation site may be originated from Kyrgyzstan, Tajikistan, Afghanistan… The air mass coming from Pakistan are not seen by the back trajectory, so we decide to exclude it from the manuscript.

Line 96-98: To my knowledge, the error analysis cannot be this simple. The error in extinction must be different from that in backscatter. Also, the error must be dependent on height and the background radiation. It should be mentioned that the Raman lidar measurement was

limited in the nighttime, if so. In addition, it would be better to have some descriptions about the advantage of using rotational Raman instead of vibrational Raman at 532 nm.

Answer: A sentence presenting the advantage of rotational Raman channel has been added and one reference paper has been given. That measurements were made in nighttime has been added in the manuscript. The error estimate is presented in the appendix <Hu et al. 2019: Long-range-transported Canadian smoke plumes in the lower stratosphere over northern France>, so it is not repeated in this paper. The error is height dependent but here we selected typical values at a certain vertical level, calculated the error and then apply it to all the vertical levels. The 15% of error is a conservative value derived with 10% of error in the backscattering coefficient, volume depolarization ratio and 200-300% in the molecular depolarization ratio. We re-calculated the error more carefully and find that in some cases, the error at 355 nm exceeds 15%, for example, Case 3. The errors in the upper layer and lower in Case 3 and 4 are calculated separately. Two examples of the calculated errors are shown in the following tables:

Case 1:

| Wavelength (nm) | R | VLDR | MDR | E_R | E_VLDR | E_MDR | PLDR | E_PLDR |
|---|---|---|---|---|---|---|---|---|
| 355 | 2.6 | 0.19 | 0.015 | 10% | 10% | 200% | 0.33 | 15% |
| 532 | 9.80 | 0.31 | 0.020 | 10% | 10% | 300% | 0.36 | 11% |

Case 3:

| Wavelength (nm) | R | VLDR | MDR | E_R | E_VLDR | E_MDR | PLDR | E_PLDR | |
|---|---|---|---|---|---|---|---|---|---|
| 355 | 1.7 | 0.83 | 0.015 | 10% | 10% | 200% | 0.21 | 21% | Upper layer |
| 532 | 2.88 | 0.16 | 0.010 | 10% | 10% | 200% | 0.24 | 13% | |
| 1064 | 10.0 | 0.23 | 0.020 | 20% | 10% | 300% | 0.26 | 11% | |
| 355 | 1.64 | 0.11 | 0.015 | 10% | 10% | 200% | 0.30 | 24% | Lower layer |
| 532 | 4.58 | 0.25 | 0.010 | 10% | 10% | 200% | 0.34 | 12% | |
| 1064 | 28.0 | 0.30 | 0.012 | 20% | 10% | 300% | 0.31 | 10% | |

Figure 3: The periods of Case1, 2, 3 and 4 should be indicated in Figure 3.
Answer: Corrected.

Figure 5: Case1, 2, 3 and 4 should be indicated in Figure 5.
Answer: They were indicated in the caption of Figure 5, so we think it is not necessary to be indicated on the figure.

Figure 3: Legend "500 nm" should be AOD (500 nm).

Answer:  Corrected.

Line 166-168: The backscatter coefficient at 1064 nm below 1800 m should be indicated in Fig. 6.

Answer:   On 09 April 2019, the aerosol content was very high, so the signal at 1064 nm is not useable because of signal distortion. This is the reason why it was not plotted in Figure 6. The explanation has been given in the manuscript.

Line 169-170: "EAE" and "BAE" are not defined.

Answer:  Thanks. It has been corrected.

Line 183-187: Is the description consistent with Figure 3?

*L182-187: "… limit of the sun/sky photometer, so the AERONET and SONET retrieval can not be applied. A large and intense plume was first detected in the morning of 23 April 2019 (Figure 4). And on 24 April, a hot spot of UVAI appeared over the observation site. The daily average of AOD is 3.63 and Ångström exponent is about -0.01, according to the daytime sun/sky photometer measurements… "*

Answer:  Yes, it is consistent. I am not sure what inconsistencies you have observed in this paragraph. I guess maybe you mean the values of AOD and AE? The values we mentioned in this paragraph are daily averaged values, not the instantaneous values in Figure 3. If you were wondering why we say "an intense plume was detected on 23 April", but that was not reflected by Figure 3, the answer is that this plume was not over our observation site. I hope I got your question.

Line 227-228: What is the "clear evidence of polluted dust"?

Answer:  It is the decrease of PLDRs and increase of EAE, which indicates the occurrence of fine particles and particles with more spherical shapes.  The increase of BAE also corroborates that aerosols above 2200 m are not the same with those below 2000 m. You might want to point out that pollution may not be the only cause, the deposition of coarse-mode and giant particles could also lead to this effect. We agree, the manuscript has been improved with taking into account this issue.

Line 229-232: The structure at around 2500 m is interesting and should be studied further. Is the type of dust (or "polluted" dust) the same in 1000-2200 m and 2400-2800 m or different? Why relative humidity was high in 2400-2800 m?

Answer:  They are different aerosol types since signatures in PLDR, EAE and BAE are different. The WVMR is also a tracer of air mass. The relatively higher WVMR or RH at 2400-2800 m indicates that the air mass at 2400-2800 m could have different origins with the air mass at lower altitudes. This is one reason why we supposed it is polluted dust. But the increase of WVMR is not significant enough to confirm that they are definitively different air mass.

Figure 9: Captions for (c) and (d)

Answer:   The caption has been complemented.